# Crystal structures of aconitase X enzymes from bacteria and archaea provide insights into the molecular evolution of the aconitase superfamily

Seiya Watanabe [1,2,3,6]✉, Yohsuke Murase[1,6], Yasunori Watanabe[1,2,4], Yasuhiro Sakurai[5] & Kunihiko Tajima[5]

Aconitase superfamily members catalyze the homologous isomerization of specific substrates by sequential dehydration and hydration and contain a [4Fe-4S] cluster. However, monomeric and heterodimeric types of function unknown aconitase X (AcnX) have recently been characterized as a cis-3-hydroxy-L-proline dehydratase (AcnX$_{Type-I}$) and mevalonate 5-phosphate dehydratase (AcnX$_{Type-II}$), respectively. We herein elucidated the crystal structures of AcnX$_{Type-I}$ from Agrobacterium tumefaciens (AtAcnX) and AcnX$_{Type-II}$ from Thermococcus kodakarensis (TkAcnX) without a ligand and in complex with substrates. AtAcnX and TkAcnX contained the [2Fe-2S] and [3Fe-4S] clusters, respectively, conforming to UV and EPR spectroscopy analyses. The binding sites of the [Fe-S] cluster and substrate were clearlydifferent from those that were completely conserved in other aconitase enzymes; however, theoverall structural frameworks and locations of active sites were partially similar to each other.These results provide novel insights into the evolutionary scenario of the aconitase superfamilybased on the recruitment hypothesis.

[1] Department of Bioscience, Graduate School of Agriculture, Ehime University, Matsuyama, Ehime, Japan. [2] Faculty of Agriculture, Ehime University, Matsuyama, Ehime, Japan. [3] Center for Marine Environmental Studies (CMES), Ehime University, Matsuyama, Ehime, Japan. [4] Faculty of Science, Yamagata University, Yamagata, Japan. [5] Department of Molecular Chemistry, Kyoto Institute of Technology, Kyoto, Japan. [6]These authors contributed equally: Seiya Watanabe and Yohsuke Murase. ✉email: irab@agr.ehime-u.ac.jp

Aconitase (Acn) was initially described in 1937 as an "enzyme system" that catalyzes the interconversion of citrate, *cis*-aconitate, and isocitrate in mammalian mitochondria just after this activity was disengaged from the oxidative decarboxylation of isocitrate to α-ketoglutarate. The Acn superfamily currently contains four functional hydro-lyase enzymes: the archetypical Acn (EC 4.2.1.3), 2-methylcitrate dehydratase (EC 4.2.1.79; AcnD), homoaconitase (EC 4.2.1.114; HACN), and isopropylmalate isomerase (EC 4.2.1.33; IPMI) (Fig. 1a). These members have been further classified into seven phylogenetic subfamilies: (1) AcnA of bacteria[1]; (2) AcnB of bacteria[2]; (3) mitochondrial Acn (mAcn)[3]; (4) cytoplasmic Acn (cAcn) and iron regulatory protein (IRP) of mammalians[4]; (5) AcnD of bacteria[5]; (6) HACN of bacteria and archaea[6,7]; (7) IPMI of bacteria, archaea, and fungi[8–10]; and (8) function unknown

aconitase X (AcnX) of bacteria, archaea, and fungi[11] (Fig. 1c). They commonly consist of four domains (fragments), and are classified into three distinct architectural variants; domain 4 is located at the C terminus (mAcn, cAcn/IRP, and AcnA: 1-2-3-linker-4) or N terminus (AcnB, fungal IPMI, and bacterial and fungal $\text{AcnX}_{\text{Type-I}}$: 4-linker-1-2-3) of a single polypeptide and exists as a separate subunit (HACN, bacterial IPMI, and archaeal and bacterial $\text{AcnX}_{\text{Type-II}}$: 1-2-3, 4).

Four functional enzymes, except for AcnX (referred to as "other Acn enzymes"), catalyze the homologous stereospecific isomerization of α- to β-hydroxyl acids by sequential dehydration and hydration (anti-elimination/addition) similar to Acn, and the structure of each domain strongly resembles its mAcn counterpart; however, their physiological roles differ clearly (Fig. 1a). Regarding their catalytic mechanisms (see Fig. 6c), the alkoxide

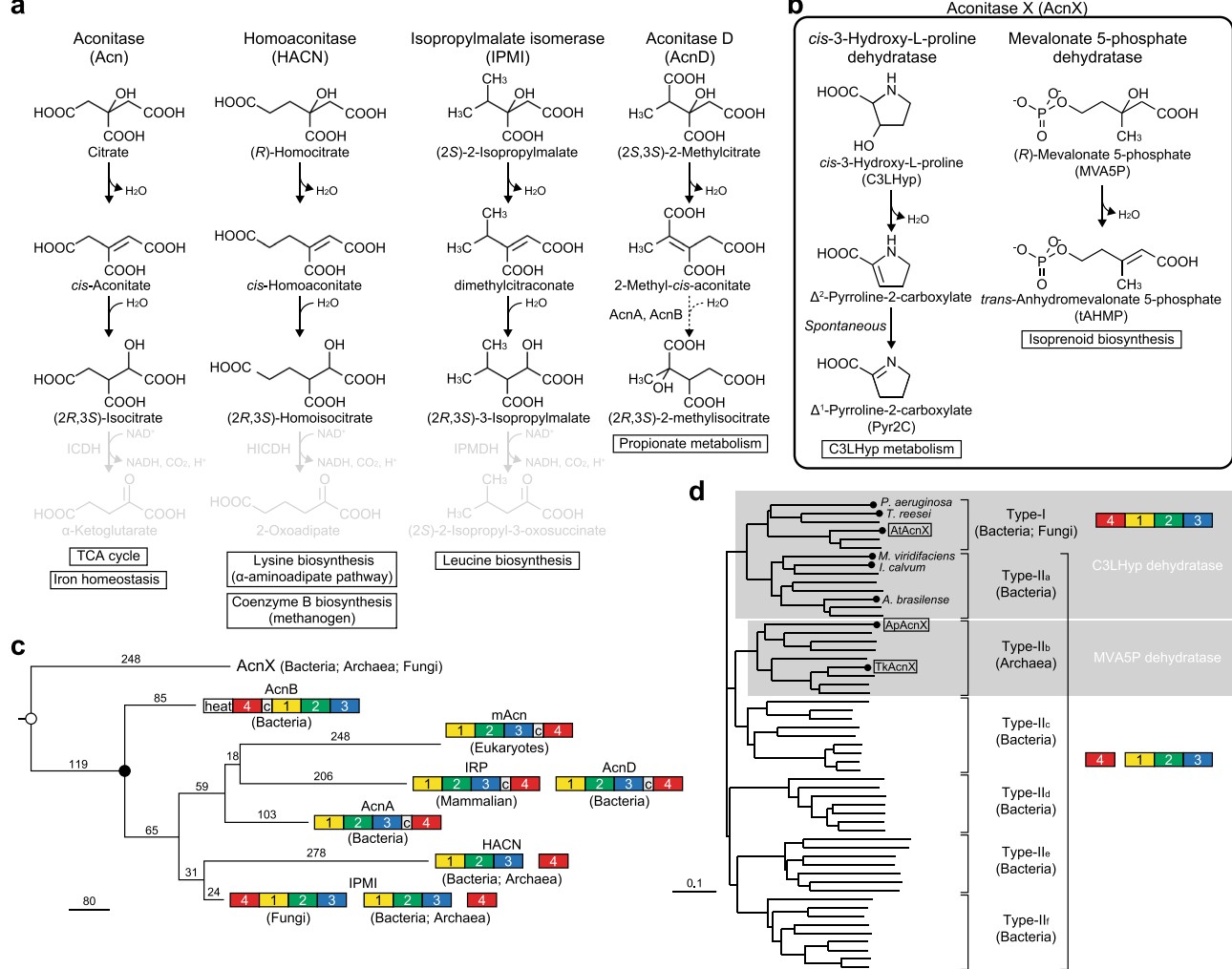

**Fig. 1 Aconitase superfamily members.** Schematic reactions of four known functional members of the aconitase superfamily (**a**) and two functional members of the AcnX subfamily (**b**). Each physiological role is in a box. In propionate metabolism, AcnD catalyzes only the dehydration of (2*S*,3*S*)-2-methylcitrate to 2-methyl-*cis*-acotinate, which is subsequently metabolized to (2*S*,3*R*)-2-methylisocitrate by either AcnA and AcnB. **c** Phylogenetic tree of the aconitase superfamily. A linear representation of the sequential domain arrangement of eight phylogenetic subfamilies is also included. "c" is the connector domain and "heat" is a protein–protein interaction domain found only in AcnB[2]. This phylogenetic tree was constructed based on the sequence identity of domain 4, in the National Center for Biotechnology Information (NCBI). The closed circle indicates the common ancestor for other aconitase enzymes that had previously been proposed, which contains the [4Fe-4S] cluster. The open circle indicates the common ancestor for the aconitase superfamily that is proposed in the present study. **d** Phylogenetic tree of the AcnX subfamily based on sequence similarity. The AcnX subfamily is further classified into $\text{AcnX}_{\text{Type-I}}$, consisting of a single polypeptide from bacteria and fungi, and $\text{AcnX}_{\text{Type-II}}$ in a number of bacteria ($\text{AcnX}_{\text{Type-IIa}}$, $\text{AcnX}_{\text{Type-IIc}}$ $\text{AcnX}_{\text{Type-IIf}}$) and archaea ($\text{AcnX}_{\text{Type-IIb}}$), which consists of (fragmented) small and large polypeptide chains. Among them, $\text{AcnX}_{\text{Type-I}}$ and $\text{AcnX}_{\text{Type-IIa}}$, and $\text{AcnX}_{\text{Type-IIb}}$ correspond to C3LHyp dehydratase and MVA5P dehydratase, respectively, in Fig. 1b. The circles at the end of each branch are the enzymes that have been functionally characterized. The large subunit of $\text{AcnX}_{\text{Type-II}}$ was used for this phylogenetic analysis. A more-detailed comparison is shown in Supplementary Fig. 8.

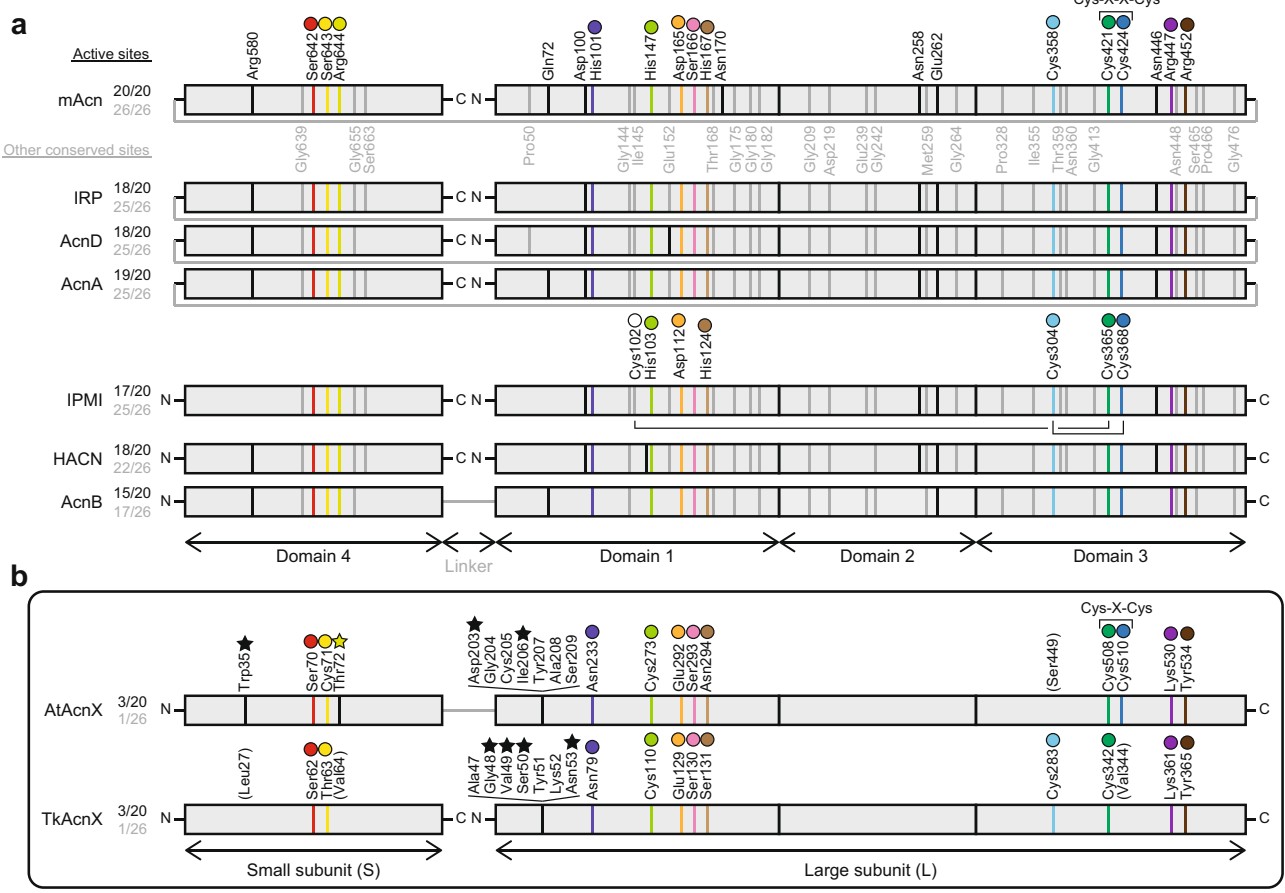

**Fig. 2 Comparative sequence analysis of aconitase superfamily enzymes. a** Other aconitase enzymes. Black and gray letters indicate 20 of the active site residues that are completely conserved in all aconitase enzymes, except for AcnX, and 26 of the fully conserved residues that have not been identified as part of the active site, respectively[13]. Active sites with the same colored circles are located at equivalent (or close) positions between other aconitase enzymes and AcnX. The open circle in IPMI is the fourth cysteine residue near the [4Fe-4S] cluster (see Fig. 4i). **b** AcnX. The amino-acid residues with black stars indicate specific active sites for C3LHyp dehydratase and MVA5P dehydratase. A more-detailed structural-based alignment is shown in Supplementary Fig. 2.

form of a serine residue side chain (Ser642 in mAcn) abstracts a proton from C2 of the substrate as a general base, and the [4Fe-4S] cluster acts as a Lewis acid. Three out of four Fe atoms in this metallic center coordinate to each cysteine residue (Cys358, Cys421, and Cys424)[12]. Of the 20 active sites, residues including these serine and cysteine residues assigned in mAcn, >15 sites are identical in IRP, AcnD, AcnA, IPMI, HACN, and AcnB: there are also another 26 completely conserved residues[13] (Fig. 2a). Therefore, the Acn superfamily (enzymes) is a typical example that is suitable for the recruitment hypothesis of enzyme evolution proposed by Jensen[14]; the gene duplication of multi-specific enzymes, followed by the narrowing of substrate specificity. Based on our knowledge of Acn, the nature of the [4Fe-4S] cluster has been revealed in detail using highly discriminating spectroscopic tools, such as electron paramagnetic resonance (EPR), Mössbauer (MB), and electron-nuclear double resonance (1978~), and the subsequent elucidation of the crystal structure (1989~), and the common ancestor of this protein superfamily appears to have already contained the [4Fe-4S] cluster (closed circle in Fig. 1c)[12].

In contrast, AcnX (subfamily) was initially discovered by a comparative analysis of archaeal genomes in 2003[11]. Although the secondary structural elements of other Acn enzymes may also be conserved in this subfamily, sequence similarities between them are insignificant (10%> identity); the majority of active sites assigned in mAcn had no readily detectable counterpart in AcnX (Fig. 2b). In 2016[15], AcnX proteins from bacteria and fungi were

functionally characterized as a *cis*-3-hydroxy-L-proline (C3LHyp) dehydratase. This enzyme catalyzes the dehydration of C3LHyp to $\Delta^1$-pyrroline-2-carboxylate via a $\Delta^2$-pyrroline-2-carboxylate intermediate, which is subsequently converted to L-proline by the reductase (Fig. 1b); this type of AcnX corresponds to $AcnX_{Type-I}$, described above and is the first functional annotation of the AcnX subfamily. The encoding gene is often located within the bacterial gene cluster involved in L-hydroxyproline metabolism, such as *trans*-4-hydroxy-L-proline and *trans*-3-hydroxy-L-proline (Supplementary Fig. 1); to the best of our knowledge, C3LHyp has only been detected as one of the building blocks of telomycin, a peptide antibiotic produced by *Streptomyces canus* C159[16]. This is a good example of the gene context in the genome being more helpful for estimating the potential substrate of a hypothetical protein than amino-acid sequence similarities. Furthermore, based on EPR and site-directed mutagenic analyses, a mononuclear Fe(III) center may be coordinated with one glutamate and two cysteine residues, (with the orange, light-green, and green circles in Fig. 2b, respectively).

On the other hand, another type of AcnX from hyperthermophilic archaea ($AcnX_{Type-II}$) was functionally characterized as a mevalonate 5-phosphate (MVA5P) dehydratase in 2018[17]. This enzyme is involved in the modified mevalonate pathway and catalyzes the dehydration of MVA5P to the previously unrecognized metabolite, *trans*-anhydromevalonate 5-phosphate (tAHMP), which is then converted to isopentenyl phosphate by

the novel decarboxylase (Fig. 1b). Furthermore, the purified enzyme is brown in color and contains an (unidentified) [Fe-S] cluster. These findings were unexpected for the following reasons: (1) MVA5P is structurally different to C3LHyp; (2) the putative [Fe-S] cluster-binding sites of AcnX$_{Type-I}$ are incompletely conserved in AcnX$_{Type-II}$ (see below); and (3) the physiological role of AcnX$_{Type-II}$ is not related to that of AcnX$_{Type-I}$.

We herein report for the first time the crystal structures of AcnX$_{Type-I}$ from *Agrobacterium tumefaciens* (AtAcnX) and AcnX$_{Type-II}$ from *Thermococcus kodakarensis* (TkAcnX). The structural frameworks of each domain (1–4) of AtAcnX and TkAcnX were similar to their counterparts of other Acn enzymes. AtAcnX and TkAcnX had the [2Fe-2S] and [3Fe-4S] clusters, respectively, conforming to UV and EPR spectroscopy analyses. Their [Fe-S] cluster-binding sites consisting of three cysteine residues were clearly different not only between them but also from other Acn enzymes. Furthermore, in the crystal structures in complex with the substrate, the common backbones of C3LHyp and MVA5P were recognized by homologous amino-acid residues. These active sites were located at equivalent (or close) positions to other Acn enzymes, whereas amino-acid residues were not conserved. A novel evolutionary scenario of the Acn superfamily that differs from the simple hypothesis by Jensen is also discussed.

## Results and discussion

### Overall structural description of AtAcnX.

Monomeric AtAcnX (~59 kDa) was expressed in *Escherichia coli* cells and purified through two chromatographic steps (Fig. 3a). The purified enzyme was brown in color, indicating the presence of cofactor(s). The crystal structure was elucidated by the single-wavelength anomalous dispersion (SAD) method using the selenomethionine (SeMet)-substituted crystal, and subsequently refined to a resolution of 1.6 Å against the data set from the native crystal (Fig. 3b). The three molecules in the asymmetric unit were essentially identical, being superimposable with a root-mean-square deviation (r.m.s.d.) of 0.4–0.7 Å over 552–558 Cα atoms. Data collection and refinement statistics are summarized in Table 1.

As described in the "Introduction", the primary structure of AcnX has also been proposed to consist of four domains, similar to other Acn enzymes, whereas their divisions on the primary structure are unclear. Alternatively, we attempted to separately superimpose each domain of mAcn onto the crystal structure of AtAcnX, and found that domains 1–4 of AtAcnX each consisted of 157–336, 337–435, 436–569, and 1–149 residues in the order of 4-1-2-3, respectively, from the N to C terminus (Supplementary Fig. 2). Among them, domains 1, 2, and 3 each commonly contained a central parallel β-sheet linked by α-helices, similar to the nucleotide-binding domains of dehydrogenases (Fig. 3e). On the other hand, the typical structural motif of domain 4 was a β-barrel created by eight β-strands, and two out of the three α-helices were packed at one side of this motif.

The structure of each domain of AtAcnX may also be superimposed on its counterpart of other Acn enzymes, except mAcn (Supplementary Fig. 3). Although their r.m.s.d. values were relatively high and sequence identities were very poor, the closest similarities appeared to be commonly found in domain 3. In comparisons with mAcn, all differences in the folding of AtAcnX were deletions, owing to which the surface area decreased by 23%; 19632 and 25655 Å$^2$, respectively (Fig. 3d). The deletion of the segment that joined the C terminus of domains 1-2-3 and the N terminus of domain 4 of mAcn was the largest (black); the N terminus of domains 1–3 and the C terminus of domain 4 of AtAcnX was connected by a short linker corresponding to

150–156 residues. Other deletions were located at the N terminus of domain 1 (gold) and domain 3 (cyan) outside of the β-barrel motif, and the C terminus of domain 4 (pink) and domain 2 (light-green), which were distant from the active site of mAcn.

### Binding of the [2Fe-2S] cluster in AtAcnX.

AtAcnX crystals were brown in color, similar to the purified protein, strongly indicating the maintenance of cofactors (photo in Fig. 3b). The electron density map suggested that this enzyme had a planar [2Fe-2S] cluster (gray mesh in Fig. 4a) located within a solvent-filled cleft in the protein structure (Fig. 3b). Furthermore, the anomalous difference Fourier map showed two clear peaks within this electron density map, confirming the presence of this type of metallic center (blue mesh in Fig. 4a). Fe2 and Fe1 atoms were tetrahedrally coordinated by two bridging sulfide ions and two cysteine residues (Cys273 and Cys508) and by two bridging sulfide ions, one water molecule, and one cysteine (Cys510), respectively (Fig. 4f). The water molecule further formed hydrogen bonds with the side chains of Glu292 and Lys530.

Among other typical [2Fe-2S] cluster-containing proteins, such as plant/vertebrate-type ferredoxin, Rieske proteins, and thioredoxin-like ferredoxin, the metallic center has been shown to commonly participate in the catalysis of a redox reaction, whereas both iron atoms were ligated to the protein by four cysteine residues in most cases[18]. Therefore, the scaffold in the [2Fe-2S] cluster coordination of AtAcnX is very unique. Similar examples were recently reported in the crystal structures of L-arabinonate dehydratase (EC 4.2.1.25; AraC)[19], D-xylonate dehydratase (EC 4.2.1.82)[20], and dihydroxy acid dehydratase (EC 4.2.1.9)[21] from bacteria, which commonly belong to the ILVD/EDD (but not Acn) protein superfamily. On the other hand, Acn is an uncommon [4Fe-4S] cluster-containing enzyme. Only the Fe1, Fe2, and Fe3 atoms showed normal ligation to cysteine residues (Cys358, Cys421, and Cys424 in mAcn), whereas the fourth Fe4 atom formed a hydrogen bond network with Asp165, His167, and His147 (Fig. 4h)[12].

The superpositions of the overall crystal structures showed that the [2Fe-2S] cluster in AtAcnX was roughly located at a similar position to the [4Fe-4S] cluster in mAcn (Fig. 3b, d and 4f, h). Among the [Fe-S] cluster-binding sites, only two adjacent Cys508 (green) and Cys510 (blue) in AtAcnX were located at equivalent (or close) positions to Cys421 and Cys424 in mAcn; the characteristic motifs of Cys-X-Cys and Cys-X-X-Cys were often detected in other [Fe-S] cluster-containing proteins (Fig. 2)[18]. On the other hand, AtAcnX possessed Cys273 (yellow-green) at an equivalent position to His147 in mAcn, whereas no homologous cysteine residues to Cys358 were found in mAcn. The hydrogen bond network via a water molecule that bound to the [Fe-S] cluster was also clearly different between AtAcnX and mAcn, although Glu292 (orange) in AtAcnX played the same role as sequentially homologous Asp165 in mAcn.

### UV and EPR spectroscopy analyses of the [2Fe-2S] cluster in AcnX$_{Type-I}$.

Based on these structural insights, we elucidated the nature of the [Fe-S] cluster in AcnX$_{Type-I}$ enzymes from not only *A. tumefaciens*, but also *Pseudomonas* sp. NBRC 111117 (PsAcnX) using UV and EPR spectroscopy, together with (originally prepared) AraC (from *Herbaspirillum huttiense*; ref. [22]). PsAcnX (initially prepared for crystallization; Fig. 3a) shows 47% sequence identity with AtAcnX (Supplementary Fig. 4) and functions as a C3LHyp dehydratase. In a previous study, signals around $g = ~4.0$ were observed for the mononuclear $S = 5/2$ Fe(III) (see Fig. 5b), but the low average $g$ value ($g_{aver}$), characteristic of Rieske and/or ferredoxin proteins, was not noted, by which we assumed that AtAcnX contains an iron ion[15].

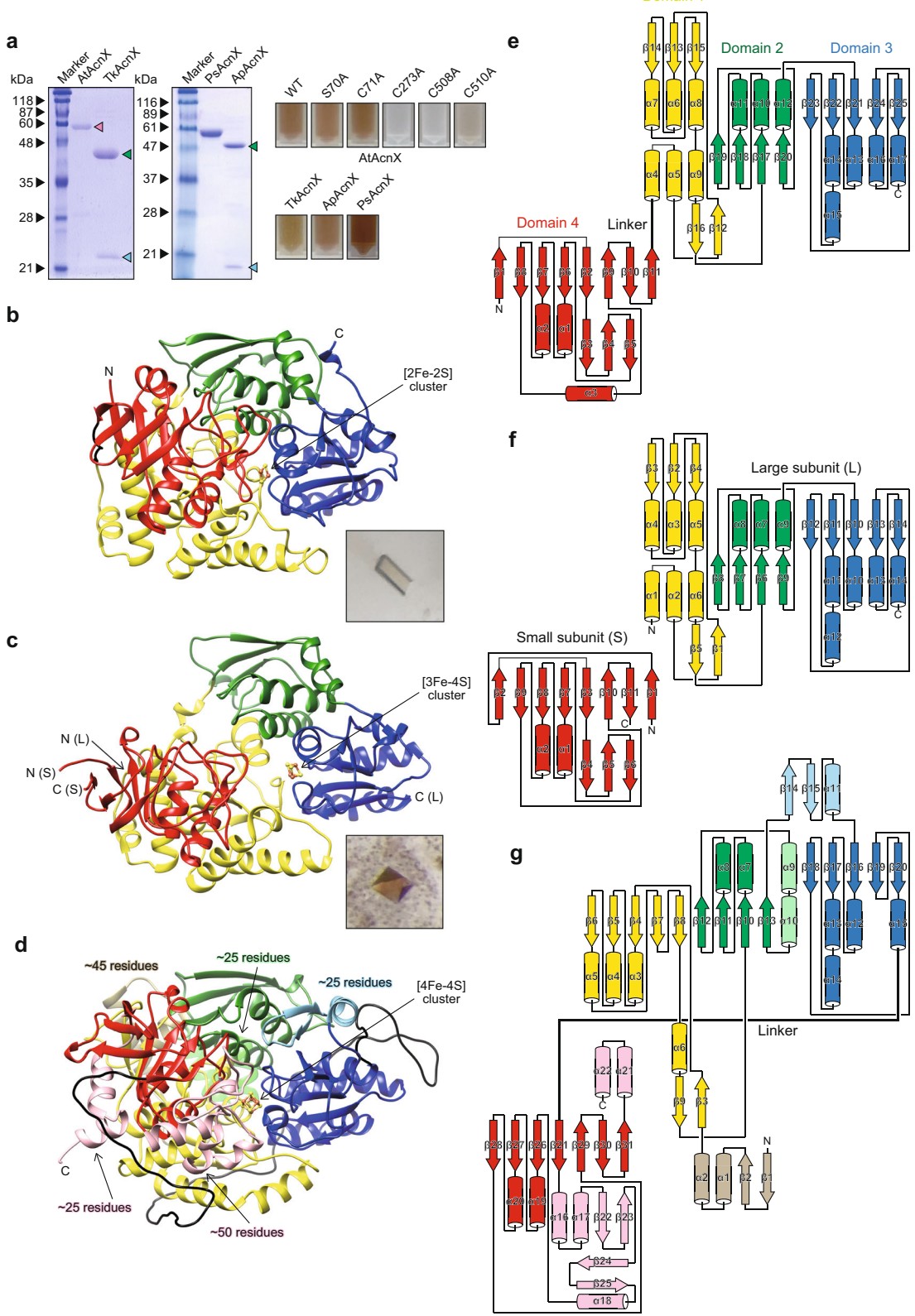

Alternatively, in this study, the (more concentrated) enzyme(s) was freshly prepared under partial anaerobic conditions.

The UV visible absorption spectra of AtAcnX, PsAcnX, and AraC show peaks at 325, 410–420, 450–460, and 520–540 nm arising from S → Fe(III) charge transfer bands (Fig. 5a, c, e). Their EPR spectra, as isolated, are essentially featureless,

consistent with the [Fe-S] center in its oxidized form (blue line in Fig. 5b, d, f) with the anti-ferromagnetically coupled [Fe(III)Fe (III)], forming a diamagnetic electron spin $S = 0$ state[23]. When the enzymes are reduced with sodium dithionite ($Na_2S_2O_4$), the resulting spectra (at 20 K) have a $g_{aver}$ value of <2.0, and are consistent with a [Fe(II)Fe(III)] $S = 1/2$ species (red line in

**Fig. 3 Overall crystal structures of AcnX. a** SDS-PAGE (10 μg in a 15% (w/v) gel) and a photograph (~20 mg/ml) of purified enzymes. AtAcnX (and PsAcnX) consists of a single polypeptide (~59 kDa; pink triangle), whereas TkAcnX (and ApAcnX) consists of small (~24 kDa; light-blue triangle) and large subunits (~42 kDa; green triangle). PsAcnX and ApAcnX were originally prepared as other AcnX$_{Type-I}$ and AcnX$_{Type-II}$ enzymes, respectively. Ribbon (**b**, **c**, **d**) and topological diagrams (**e**, **f**, **g**) of AtAcnX (**b**, **e**), TkAcnX (**c**, **f**), and mAcn (**d**, **g**; PDB ID, 6ACN) in apo-forms. These molecules commonly contain the indicated [Fe-S] cluster (represented as a ball-and-stick model), and consist of four distinct domains 1 (yellow), 2 (green), 3 (blue), and 4 (red) and a linker region (black). The small subunit of TkAcnX corresponds to fragmented domain 4. In comparisons with AcnX, several large insertions in domains 1–4 of mAcn were colored in ocher, light-green, light-blue, and pink, respectively. The superposition of each domain between AtAcnX and mAcn is shown in Supplementary Fig. 3.

Fig. 5b, d, f). Collectively, these results of the spectroscopic analysis suggest the presence of a [2Fe-2S] cluster in AcnX$_{Type-I}$ enzymes.

A six-line spectrum around $g = ~2.0$ of AtAcnX and PsAcnX, both isolated and reduced, was characteristic for Mn$^{2+}$[24]. In mAcn, the fourth Fe4 atom in the [4Fe-4S] cluster may be substituted to Mn$^{2+}$, and there are secondary binding sites for Mn$^{2+}$, except the [Fe-S] cluster[12]. Therefore, there is one possibility in which (the purified) AtAcnX possesses Mn$^{2+}$ at the equivalent position to Mg$^{2+}$ in the structure of the apo-form (right panel in Fig. 6e), and this substitution is owing to the crystallization solution containing highly concentrated MgCl$_2$.

**Binding of C3LHyp to the active site of AtAcnX.** Ser70 in AtAcnX may be sequentially homologous to Ser642 in mAcn, which functions as a general base (Fig. 2); the purified S70A mutant is almost inactive (Supplementary Fig. 5) and brown in color, similar to the wild-type enzyme (Fig. 3a). Therefore, to obtain a more-detailed understanding of substrate binding, we elucidated the crystal structure of the S70A mutant of AtAcnX in complex with C3LHyp at a resolution of 2.0 Å using a co-crystallizing method; this structure had a similar conformation to that of the wild-type enzyme in the apo-form, with an r.m.s.d. value of 0.4 Å for 559 Cα atoms (Fig. 6a). This mutant enzyme had the same [2Fe-2S] cluster as the wild-type enzyme (Fig. 4b), and additional electron density around the [2Fe-2S] cluster was clearly modeled as C3LHyp (Fig. 6c).

The carboxyl group of bound C3LHyp formed hydrogen bond(s) and/or salt bridge(s) with a water molecule (Wat16), the side chains of Ser293 and Lys530, and the main chain nitrogen atoms of Cys71 and Ser293 (Fig. 6g). This Wat16 also interacted with the side chains of Tyr534 and Ser293. The nitrogen atom of C3LHyp made a hydrogen bond and/or salt bridge with the side chains of Thr72 and Asp203, respectively (Fig. 6h). The C3-OH group interacted with the Fe1 atom and side chains of Glu292 and Lys530 (Fig. 4a, b). The side chains of Asn233 and Asn294 formed hydrogen bonds with Glu292. Collectively, two oxygen atoms, nitrogen, and C3 atoms, C3-OH group, and Wat16 in the carboxyl group of C3LHyp positionally correspond to Wat13, 83, 448, 42, and 3 and Mg$^{2+}$ in the apo structure (right panel of Fig. 6e). The hydrophobic side chains of Trp35 and Ile206 interacted with the C4-C5 backbone of the pyrrolidine ring of C3LHyp, which may explain the extremely poor utilization of 2,3-cis-3,4-cis-3,4-dihydroxy-L-proline (C3LHyp with the additional hydroxyl group on C4; Supplementary Fig. 1) as a substrate[15].

Among these active sites including the ligands for the [2Fe-2S] cluster, each alanine mutant of Ser70, Cys71, Asp203, Cys273, Cys508, Cys510, Lys530, and Tyr534 in a previous study[15] and the present study were (almost) inactive (Supplementary Fig. 5). Four W35A, T72A, I206A, and S293A mutants were subjected to further kinetic analyses with C3LHyp, and the parameters obtained are listed in Supplementary Table 1. Their $k_{cat}/K_m$ values were also reduced by 1~4 orders of magnitude from the wild-type enzyme, and were commonly caused by low $k_{cat}$ values (particularly the W35A mutant). Furthermore, the purified

proteins of C273A, C508A, and C510A (but not C71A) were colorless (Fig. 3a). Collectively, the results of these site-directed mutagenic analyses were consistent with the structural insights. Schematic diagrams of the interactions between C3LHyp and nearby amino-acid residues are also shown in Fig. 7a.

Ser70 was clearly the first candidate as a general base because when the structure of the apo-form of the wild type was superimposed onto that of the holo-form of the S70A mutant, the side chain of Ser70 was close to C2 of C3LHyp (a distance of 2.7 Å). Therefore, we propose that the dehydration reaction begins by the abstraction of a proton from C2 of C3LHyp by the alkoxide form of the Ser70 side chain, similar to mAcn (Ser642) (Fig. 7a). The Ser70 side chain forms hydrogen bonds with the main chain nitrogen atoms of Cys71 and Thr72 (right panel of Fig. 6e), by which nucleophilicity may be increased. The Fe2 atom, which exists in the [2Fe-2S] cluster in the Fe$^{3+}$ oxidation state (Fig. 5b, d), acts as a Lewis acid that accepts the electron pair from the leaving hydroxyl group on C3. The carbanion intermediate formed may be stabilized by the positively charged side chain of Lys530, which also has a role in protonating the hydroxyl and yielding the products $\Delta^2$-pyrroline-2-carboxylate and H$_2$O; the former is spontaneously converted to $\Delta^1$-pyrroline-2-carboxylate (Fig. 1b). It is important to note that Lys530 is not sequentially or structurally homologous to His101 in mAcn with the same function[12]. These orientations of active sites may also explain why trans-3-hydroxy-L-proline, the C3 isomer of C3LHyp (Supplementary Fig. 1), is not utilized for AtAcnX as a substrate.

**Overall structural description of TkAcnX.** As described in the "Introduction", AcnX$_{Type-IIb}$ from hyperthermophilic archaea (Aeropyrum pernix; ApAcnX) has been functionally characterized as an MVA5P dehydratase; however, some differences from AcnX$_{Type-I}$ have been reported[17]. Therefore, to elucidate the catalytic mechanism at the atomic level, we performed a crystallographic analysis of AcnX$_{Type-IIb}$ from TkAcnX; the small and large subunits (TkAcnX$_S$ and TkAcnX$_L$) showed sequence identities of 43% and 37% with those of ApAcnX, respectively (Supplementary Fig. 6). Genes encoding each subunit were co-expressed in E. coli cells. Not only TkAcnX$_S$ with a (His)$_6$-tag at N-termini (~24 kDa) but also TkAcnX$_L$ (~42 kDa) were co-purified using nickel-chelating affinity chromatography (Fig. 3a); the (His)$_6$-tag was proteolytically removed from the recombinant protein before crystallization. The native molecular mass, estimated by gel filtration, was ~70 kDa, indicating that this protein molecule may be composed of a heterodimeric structure. In spite of the same brown color as (originally prepared) ApAcnX (Fig. 3a), purified TkAcnX did not exhibit MVA5P dehydratase activity.

The crystal structure of TkAcnX in the apo-form was elucidated at 3.4 Å by molecular replacement using the coordinate set for the structure of AtAcnX in the apo-form (Fig. 3c); recombinant ApAcnX was also originally prepared, but not crystallized (Fig. 3a and Supplementary methods). The crystallographic asymmetric unit contained four TkAcnX molecules consisting of four chains of TkAcnX$_S$ (chains B, D, F, and H) and

**Table 1 Data collection and refinement statistics.**

| | AtAcnX | SeMet-labeled AtAcnX | AtAcnX S70A mutant with C3LHyp | AtAcnX S449C/C510V mutant | TkAcnX | TkAcnX with MVA5P |
|---|---|---|---|---|---|---|
| PDB code | 7CNP | - | 7CNQ | 7D2R | 7CNR | 7CNS |
| *Data collection* | | | | | | |
| Beamline | BL38B1 (SPring-8) | BL38B1 (SPring-8) | BL41XU (SPring-8) | BL45XU (SPring-8) | BL38B1 (SPring-8) | BL41XU (SPring-8) |
| Space group | $P2_1$ | $P2_1$ | $P2_1$ | $P2_1$ | $P4_32_12$ | $P4_22_12$ |
| *Cell dimensions* | | | | | | |
| a, b, c (Å) | 73.13, 73.15, 133.60 | 73.15, 73.12, 133.37 | 73.25, 73.45, 175.18 | 54.76, 72.68, 58.99 | 141.72, 141.72, 278.11 | 102.44, 102.44, 115.85 |
| α, β, γ (°) | 90.00, 101.39, 90.00 | 90.00, 101.38, 90.00 | 90.00, 94.13, 90.00 | 90.00, 99.68, 90.00 | 90.00, 90.00, 90.00 | 90.00, 90.00, 90.00 |
| Wavelength (Å) | 1.00000 | 0.97910 | 1.00000 | 1.00000 | 1.00000 | 1.00000 |
| Resolution range (Å) | 50.0-1.60 (1.70-1.60) | 50.0-1.80 (1.91-1.80) | 50.0-2.00 (2.12-2.00) | 50.0-2.00 (2.13-2.00) | 50.0-3.39 (3.59-3.39) | 50.0-1.90 (2.02-1.90) |
| $R_{merge}$ | 0.071 (0.378) | 0.136 (1.336) | 0.136 (0.899) | 0.080 (0.707) | 0.187 (1.263) | 0.058 (0.801) |
| $R_{meas}$ | 0.099 (0.521) | 0.144 (1.426) | 0.160 (1.054) | 0.095 (0.843) | 0.203 (1.373) | 0.068 (0.952) |
| $I/\sigma$ | 6.44 (1.83) | 10.32 (1.33) | 5.51 (1.26) | 9.67 (1.47) | 9.51 (1.39) | 11.54 (1.27) |
| Completeness (%) | 95.4 (94.4) | 97.8 (87.7) | 98.1 (96.6) | 99.6 (98.7) | 99.7 (98.5) | 98.4 (95.5) |
| Redundancy | 1.8 (1.7) | 9.9 (8.3) | 3.5 (3.7) | 3.5 (3.4) | 6.6 (6.5) | 3.5 (3.3) |
| $CC_{1/2}$ | 0.994 (0.751) | 0.999 (0.653) | 0.997 (0.721) | 0.998 (0.930) | 0.997 (0.574) | 0.999 (0.557) |
| *Refinement* | | | | | | |
| No. of refractions | 180510 | | 123114 | 59949 | 39874 | 48577 |
| $R/R_{free}$ | 0.176/0.205 | | 0.209/0.253 | 0.188/0.233 | 0.182/0.225 | 0.169/0.201 |
| *No. of atoms* | | | | | | |
| Protein | 12334 | | 16396 | 4084 | 15715 | 3948 |
| [Fe-S] cluster | 12 | | 16 | 4 | 28 | 7 |
| Ligand/ion | 3 | | 36 | 7 | - | 14 |
| Water | 2130 | | 1048 | 227 | - | 250 |
| *B-factors (Å$^2$)* | | | | | | |
| Protein | 18.4 | | 30.5 | 37.8 | 94.9 | 40.5 |
| [Fe-S] cluster | 10.6 | | 23.5 | 29.9 | 119.8 | 34.4 |
| Ligand/ion | 11.1 | | 19.5 | 37.0 | - | 37.3 |
| Water | 29.1 | | 32.1 | 38.5 | - | 44.4 |
| *r.m.s. deviations* | | | | | | |
| Bond lengths (Å) | 0.003 | | 0.004 | 0.005 | 0.004 | 0.008 |
| Bond angles (°) | 0.72 | | 0.71 | 0.75 | 0.67 | 0.93 |

Values in parentheses are for the highest-resolution shell.

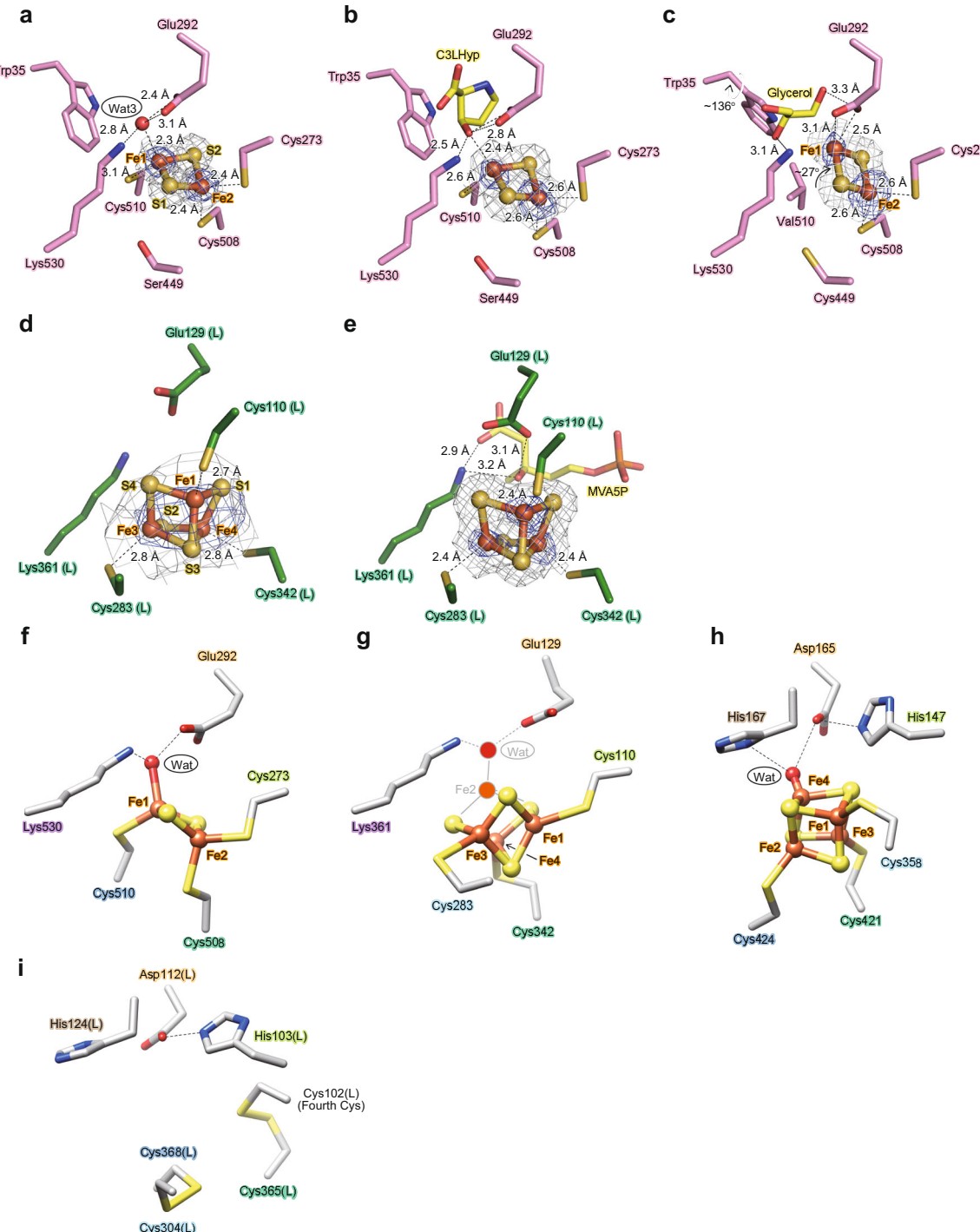

**Fig. 4 Analysis of the [Fe-S] cluster in AcnX.** The observed [2Fe-2S] cluster of AtAcnX in the apo-form (**a**) and in complex with C3LHyp (**b**) of the wild-type enzyme, and in the apo-form of the S449C/C510V mutant (**c**). The observed [3Fe-4S] cluster of TkAcnX in the apo-form (**d**) and in complex with MVA5P (**e**). $2mF_o–DF_c$ electron density maps for the [Fe-S] cluster, contoured at 1.5 σ, are shown as a gray mesh. Anomalous difference Fourier maps, contoured at the 5.0 σ (**a**), 3.0 σ (**b**), 4.0 σ (**c**), 3.0 σ (**d**), and 9.0 σ levels (**e**), indicate peaks for Fe atoms, and are shown as a blue mesh. Superposition of [Fe-S] clusters of AtAcnX (**f**), TkAcnX (**g**), mAcn (**h**; 6ACN), and IPMI (**i**; 4KP2) within its active sites, ligated to protein cysteines by Fe-S bonds. Amino-acid residues located at equivalent (or close) positions are indicated in the same color and correspond to Fig. 2. The hypothetically activated [4Fe-4S] cluster of TkAcnX, the Fe2 atom, and a water molecule (gray) appeared to be coordinated to the [3Fe-4S] cluster.

four chains of TkAcnX$_L$ (chains A, C, E, and G), the conformations of which (except for the chain B) were very similar; r.m.s.d. values on the Cα atoms were 0.36–0.42 and 0.29–0.37 Å, respectively. As expected from the primary structure, each TkAcnX$_S$ (residues 1–130) and TkAcnX$_L$ subunit (residues 1–386) superimposed well on residues 1–144 and 157–558 of AtAcnX, with r.m.s.d. values of 1.3 Å over 122 Cα atoms and 1.6 Å over 372 Cα atoms, respectively (Supplementary Fig. 2). Small differences in folding were only observed in TkAcnX$_S$; the N-terminal region formed an additional β-strand (β1), by which

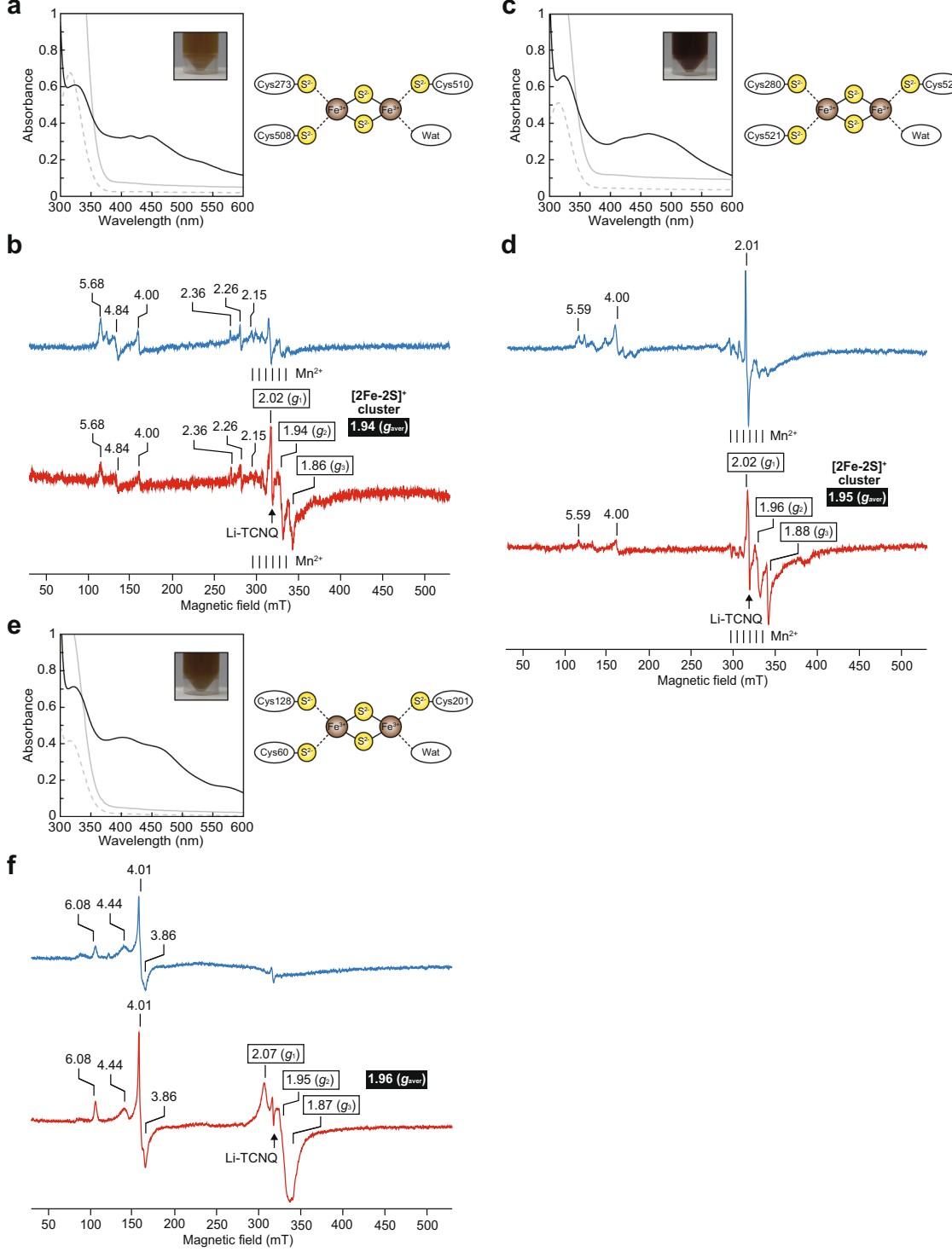

**Fig. 5 UV and EPR spectroscopy.** UV visible absorption spectra of AtAcnX (**a**), PsAcnX (**c**), and AraC (**e**) as isolated (black) and following reduction with $Na_2S_2O_4$ (gray). A fivefold diluted sample of the latter is shown as a dashed line. Inset photographs indicate the purified recombinant enzyme (~40 mg/ml). Light panels are schematic descriptions of the [2Fe-2S] cluster. EPR spectra of AtAcnX (**b**), PsAcnX (**d**), and AraC (**f**) as isolated (blue) and following reduction with $Na_2S_2O_4$ (red) at 20 K. The numbers in boxes are the $g$ values of prominent features for the [2Fe-2S] cluster. White letters in black boxes are the average $g$ values of $g_1$, $g_2$, and $g_3$.

the β-barrel was created by nine β-strands, and a shorter loop was connected between β9 and β10 instead of α3 in AtAcnX (Fig. 3f). On the other hand, the heterodimeric structure of TkAcnX showed a more "opened" conformation than that of AtAcnX (Fig. 6a, b).

**Binding of the [3Fe-4S] cluster in TkAcnX.** The electron density map and anomalous difference Fourier map suggested that TkAcnX has a cuboidal [3Fe-4S] cluster (Fig. 4d). Fe1, Fe3, and Fe4 atoms were ligated to the protein by Cys110(L), Cys342(L), and Cys283(L), respectively (the letters in parentheses indicate

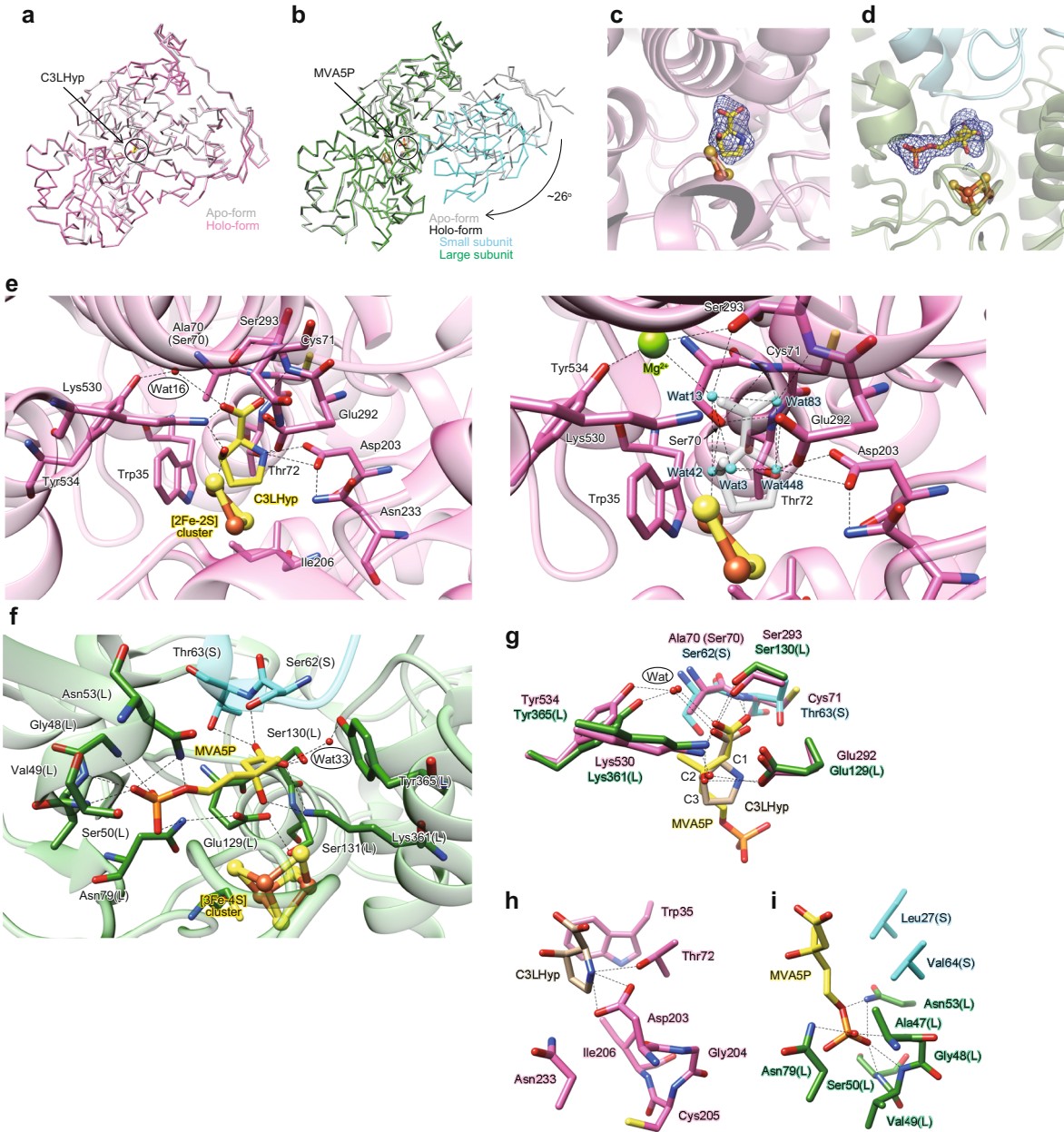

**Fig. 6 Analysis of substrate-binding sites in AcnX.** A C$_\alpha$ trace of AtAcnX (**a**) and TkAcnX (**b**) showing similarities between the apo- and holo-forms. Bound C3LHyp and MVA5P are represented as a ball-and-stick model. In TkAcnX, the holo-form showed a more closed conformation than the apo-form by rigid-body subunit movement (~26°). Electron density maps of bound C3LHyp (**c**) and MVA5P (**d**). Simulated annealing $mF_o - DF_c$ difference Fourier maps were calculated by omitting each molecule, and are shown as blue meshes countered at the 3.0 σ level. These angles were similar to those in **e** and **f**, respectively. Substrate recognition sites of AtAcnX (**e**) and TkAcnX (**f**). Active site residues are represented as stick models. The right panel of AtAcnX is the active site of the apo-form, in which C3LHyp (gray stick model) in the holo-form is superimposed. Five water molecules and one magnesium ion are represented as cyan and light-green balls, respectively. In TkAcnX, light-blue and green residues are derived from the small (S) and large subunits (L), respectively. The superposition of active sites that recognize common (**g**) and different structural backbones (**h**, **i**) between C3LHyp and MVA5P.

the location in the subunit) (Fig. 4g). Superposition to the structures of AtAcnX and mAcn showed that the [3Fe-4S] cluster was (roughly) located at equivalent positions to the [2Fe-2S] and [4Fe-4S] clusters, respectively. On the other hand, among the [Fe-S] cluster-binding sites, only Cys110(L) (yellow-green) and Cys342(L) (green) were homologous to Cys273 and Cys508 in AtAcnX, respectively, whereas Cys283(L) (aqua) corresponded to Cys358 in mAcn (Figs. 2 and 4f, g, h).

The [3Fe-4S] cluster must be derived from the [4Fe-4S] cluster unit via the loss of one iron atom. The [3Fe-4S] cluster in (inactive) mAcn may be converted to the [4Fe-4S] cluster by the

addition of ferrous ammonium sulfate anaerobically[3]. Although we attempted to reconstruct the [4Fe-4S] cluster in not only TkAcnX, also (functionally characterized) ApAcnX using similar methods, MVA5P dehydratase activity was not observed in a negative ion ESI-MS analysis; therefore, purification under anaerobic conditions appears to be crucial. The UV visible spectra of TkAcnX and ApAcnX commonly showed an extra absorption band at 410 nm (Supplementary Fig. 7a, b), which were very similar to those of inactive mAcn containing the [3Fe-4S] cluster[25] and aluminum-stressed AcnA containing the disturbed [Fe-S] cluster[26]. Furthermore, EPR spectroscopy of

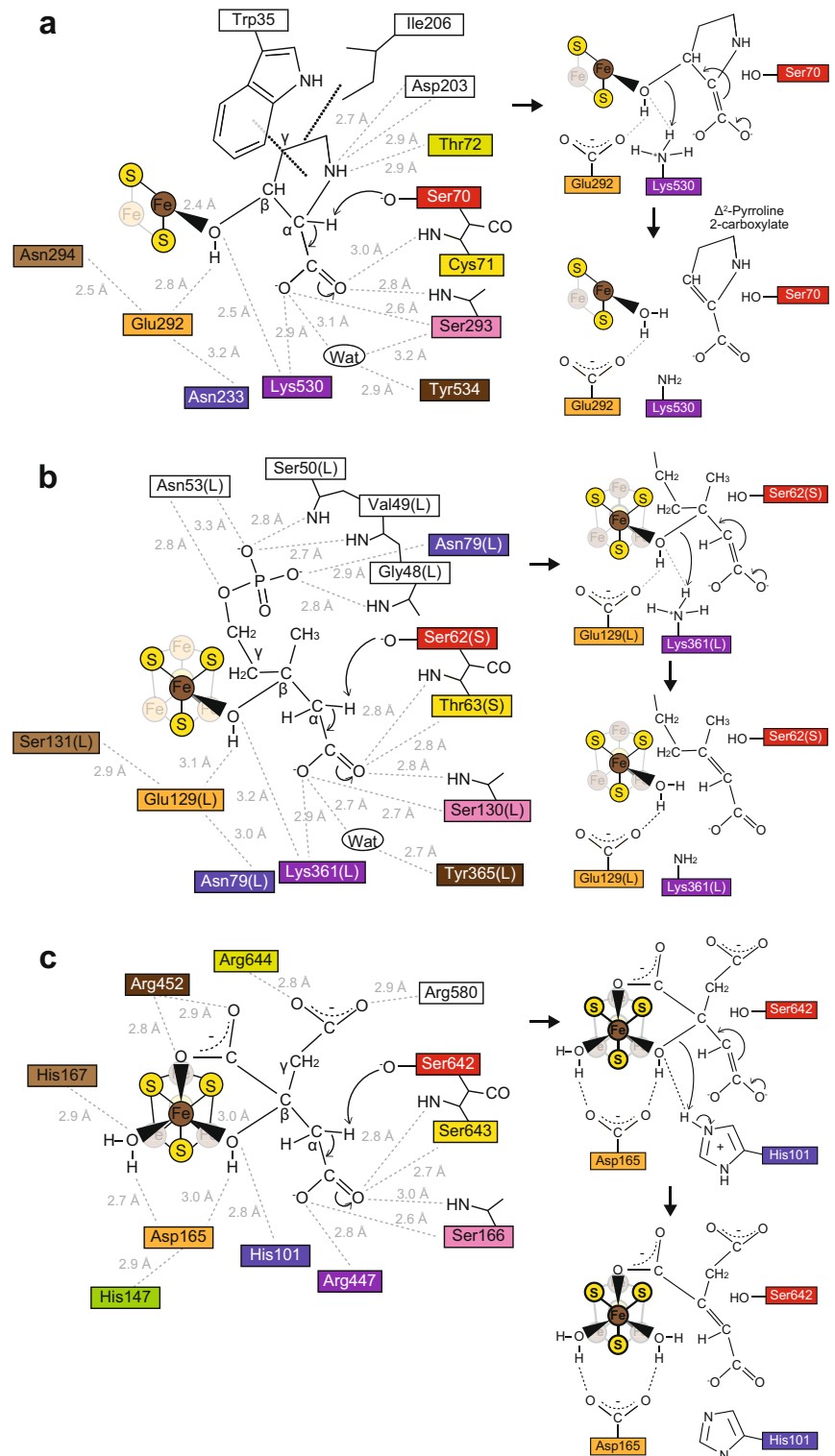

**Fig. 7 Putative catalytic mechanism.** The left panel shows schematic diagram showing interactions between the substrate and nearby amino-acid residues in AtAcnX (**a**), TkAcnX (**b**), and mAcn (**c**). Amino-acid residues that are located at equivalent (or close) positions are indicated in the same color and correspond to Fig. 2. The light panel shows a schematic representation of the reaction assuming the deprotonation of a serine residue (red) when the substrate binds and Fe-OH$_2$ is formed.

TkAcnX provided no evidence for the presence of the [4Fe-4S] cluster (Fig. 7c). Since no clear peak in void volume was observed during the gel-filtration for (aerobic) purification of TkAcnX, it is likely that the inactivation is not due to protein aggregation.

On the other hand, all binding sites for the [3Fe-4S] cluster and MVA5P in TkAcnX were completely conserved in ApAcnX (Supplementary Fig. 6), and *T. kodakarensis* possessed homologous genes (TK_RS04575 and TK_RS02520) to tAHMP

decarboxylase and its UbiX-partner from *A. pernix* (APE_RS06980 and APE_RS05555) involved in the modified mevalonate pathway[17]. Therefore, we assumed that all (active) AcnX$_{Type\ II}$ including TkAcnX and ApAcnX commonly function as an MVA5P dehydratase, and contain the [4Fe-4S] cluster as a metallic center. Although the crystal structure(s) of TkAcnX elucidated in the present study corresponded to the inactive form, in hypothetically activated TkAcnX, the Fe2 atom in the [4Fe-4S] cluster may interact with a water molecule, Glu129(L), and Lys361(L) as well as AtAcnX (Fig. 4g).

**Binding of MVA5P in the active site of TkAcnX.** To clarify substrate binding, we directly co-crystallized the wild-type enzyme of TkAcnX (inactive form) with MVA5P, and the crystal structure was elucidated at a resolution of 1.9 Å. The [3Fe-4S] cluster was located at the same position as the structure of the apo-form (Fig. 4e), and additional electron density around the [3Fe-4S] was clearly modeled as MVA5P (Fig. 6d). In the structure of the holo-form, TkAcnX$_S$ and TkAcnX$_L$ showed no large conformational changes from those in the apo-form (chains D and C); r.m.s.d. values on Cα atoms were 0.5 and 0.6 Å, respectively. On the other hand, the two subunits were closely rotated by an angle of ~26°; substrate binding resulted in a more "closed" conformation than the structure in the apo-form (Fig. 6b).

The carboxyl group of bound MVA5P made hydrogen bond(s) and/or salt bridge(s) with the side chains of Thr63(S), Ser130(L), and Lys361(L), the main chain nitrogen atoms of Thr63(S) and Ser130(L), and a water molecule (Fig. 5f). The water molecule further interacted with the side chain of Tyr365(L). The hydroxyl group of MVA5P on C3 formed hydrogen bonds with the side chains of Glu129(L) and Lys361(L), among which the former interacted with the side chains of Asn79(L) and Ser131(L). The 5-phosphate group made hydrogen bonds with the main chain nitrogen atoms of Gly48(L), Val49(L), and Ser50(L), and the side chains of Asn53(L) and Asn79(L). Among them, Asn79(L) was specific for TkAcnX, and Asn233 in AtAcnX, corresponding to Asn79(L) in TkAcnX, was not directly involved in substrate binding. On the other hand, the side chains of Thr72, Asp203, Trp35, and Ile206 in AtAcnX, responsible for interacting with the nitrogen atom or pyrrolidine ring of C3LHyp, were substituted to (non-polar) Val64(S), Ala47(L), (non-cyclic) Leu27(S), and (hydrophilic) Ser50(L) in TkAcnX, respectively.

Collectively, the (superimposed) carboxyl and C3-OH groups and the C1-C2-C3 backbones of C3LHyp and MVA5P were recognized by homologous residues between AtAcnX and TkAcnX (Fig. 6g), whereas their positions differed among specific residues (Fig. 6h, i). In addition, Ser62(S) was close to C2 of MVA5P (a distance of 3.2 Å) and superimposed onto Ser70 in AtAcnX. Therefore, the catalytic mechanism of TkAcnX may be similar to that of AtAcnX, except for the involvement of the [4Fe-4S] cluster as a Lewis acid instead of the [2Fe-2S] cluster (Fig. 7b). As MVA5P has an acyclic structure, this reaction may correspond to the first half of the Acn-like dehydration reactions, in which a 180° rotation of *cis*-aconitate occurs around an axis perpendicular to the double bond (referred to as "flip") (Fig. 1a). In propionate metabolism, (2 *S*,3 *S*)-2-methylcitrate (2-MC) is converted to (2 *R*,3 *S*)-2-methylisocitrate (2-MIC) via 2-methyl-*cis*-aconitate (2-MCA). AcnD only catalyzes the dehydration of 2-MC to 2-MCA because the flip of 2-MCA to the "2-MIC mode" may be prevented due to a steric clash of the methyl group with (unidentified) residue(s)[5]. This mechanism may also explain why *trans*-anhydromevalonate is the end product from MVA5P by TkAcnX.

**Molecular evolution of the AcnX subfamily.** The [4Fe-4S] cluster is the most widespread and biologically relevant [Fe-S] cluster; for example, proteins containing the [4Fe-4S] cluster outnumber those with the [2Fe-2S] cluster by a factor of >3[18]. The versatility and robustness of the [4Fe-4S] cluster through its greater stability and ability to nest in protein sites may also have been decisive in the early periods of protein evolution, during which the majority of basic folds presumably emerged. Therefore, one of the most important insights obtained in the present study is the discovery of a [2Fe-2S] cluster-containing enzyme in the Acn superfamily. To elucidate its evolutionary history, the S449C/C510V mutant of AtAcnX was newly designed. As this mutant corresponds to AcnX$_{Type-I}$ with the same binding sites for the [Fe-S] cluster as AcnX$_{Type-II}$ (Fig. 2b), the [4Fe-4S] cluster may ligate to Cys273, (the introduced) Cys449, and Cys508. The crystal structure of this AtAcnX mutant was successfully elucidated at a resolution of 2.0 Å (Table 1). The overall structure was very similar to that of the wild type in the apo-form, with an r.m.s.d. value of 0.3 Å for 558 Cα atoms. The $F_o-F_c$ omit map unambiguously showed that glycerol (derived from a cryoprotectant) binds to the active site and forms hydrogen bonds with the side chains of Glu292 and Lys530 (Fig. 4c). Among the active sites, the side chain of Trp35 was rotated at an angle of ~136°. This conformational change may have a critical effect in catalysis, similar to the alanine mutant (Supplementary Table 1); the S449C/C510V mutant was completely inactive.

An anomalous difference Fourier map showed that this mutant enzyme had the [2Fe-2S] cluster (Fig. 4c). On the other hand, the [2Fe-2S] cluster may fluctuate less for the following reasons: (1) the B-factor values of the Fe1, Fe2, S1, and S2 atoms (43.98, 23.19, 29.57, and 22.66, respectively, for chain A) were higher than those in the wild-type enzyme (10.49, 9.11, 10.18, and 10.49, respectively, for chain A); (2) the anomalous difference peak in the Fe1 atom was clearly smaller than that in the Fe2 atom; and (3) the B-factor value of the Fe2 atom was higher than that of the Fe1 atom. Glu292 (but not Cys449) directly ligated to the Fe1 atom, whereas Cys273 and Cys508 ligated to the Fe2 atom similar to the wild-type enzyme, by which the [2Fe-2S] cluster was closely rotated to Glu292 by an angle of ~27°. This was a very uncommon arrangement for the [Fe-S] cluster for the following reasons. The "complete" lack of the fourth ligand position left a vacant coordination site for the Fe1 atom, which differed from the apo-form of the wild-type enzyme. Similar phenomena have recently been reported for L-arabinonate dehydratase[19] and D-xylonate dehydratase[20]. Furthermore, in spite of the discovery of many [Fe-S] cluster-containing proteins, histidine, glutamine, arginine, and serine residues are also primarily implemented in a few cases in which amino-acid residue(s), except for cysteine, function as ligands in natural and engineered proteins[18]; therefore, this is the first report of the glutamate residue. Although the [Fe-S] cluster was unsuccessfully engineered in AtAcnX, these results provide important insight(s) into other AcnX-like protein(s), as described in the next section.

**Phylogenetic insights into the AcnX subfamily.** A Protein-BLAST (Basic Local Alignment Search Tool) analysis revealed that AcnX$_{Type-II}$ from bacteria and archaea are phylogenetically classified into six subclasses (a–f), among which AcnX$_{Type-IIb}$ including TkAcnX (and ApAcnX) are from archaea (Fig. 1d). Among the remaining four subfamilies, AcnX$_{Type-IIa}$ shows the highest sequence similarity to AcnX$_{Type-I}$. The enzymes from several bacteria, including *Azospirillum brasilense*, *Intrasporangium calvum*, and *Micromonospora viridifaciens*, have been functionally characterized as C3LHyp dehydratases containing an (unidentified) [Fe-S] cluster[27]. C3LHyp-binding sites in

AcnX$_{Type-I}$ were completely conserved in AcnX$_{Type-IIa}$ (Supplementary Fig. 8). These results indicate that AcnX$_{Type-I}$ evolved by the gene fusion of AcnX$_{Type-IIa}$ after the acquisition of enzyme function; the combination of two subunits of the AcnX$_{Type-IIa}$ enzyme from *A. brasilense* without a linker did not affect C3LHyp dehydratase activity. On the other hand, the molecular evolution of the [Fe-S] cluster between AcnX$_{Type-I}$ and AcnX$_{Type-IIa}$ remains unclear because the latter potentially possessed only two cysteine residues (yellow-green and green) for [Fe-S] cluster-binding sites (Supplementary Fig. 8).

Therefore, the characterization of the S449C/C510V mutant of AtAcnX, described above, is important because there is one possibility in which the highly conserved glutamate residue at an equivalent position to Glu292 in AtAcnX inherently has a role as one of the [Fe-S] cluster-binding sites in AcnX$_{Type-IIa}$; a crystallographic analysis is currently underway. Furthermore, as the remaining AcnX$_{Type-IIc}$~AcnX$_{Type-IIf}$ subclasses possessed no landmark residue as C3LHyp dehydratase or MVA5P dehydratase (with stars in Supplementary Fig. 8), their functional characterizations may also be helpful for verifying these hypotheses in more detail.

## Molecular evolution of the Acn superfamily

Structural comparisons between AcnX and other Acn enzymes allowed us to rewrite the evolutionary scenario of the Acn superfamily. The common ancestor (open circle in Fig. 1c), appearing before the previously proposed one (closed circle), had a similar structural framework. It possessed four serine (red), serine (pink), asparagine or glutamine (orange), and/or cysteine residues (green) in Fig. 2 as potentially active sites, whereas there was no [Fe-S] cluster. It is important to note that the majority of the other active sites were located at sequentially equivalent (or close) positions, whereas amino-acid residues were not conserved (Fig. 7), indicating that they had acquired each other independently. This hypothesis may be particularly strengthened by TkAcnX (AcnX$_{Type-IIb}$) containing the [4Fe-4S] cluster and utilizing the acyclic substrate, in contrast to AtAcnX (AcnX$_{Type-I}$); these binding types are not similar to those of other Acn enzymes.

The β-decarboxylating dehydrogenases, ICDH, HICDH, and IPMDH, included in the TCA cycle and the L-lysine and -leucine biosynthetic pathways belong to a single protein superfamily, whose relationships are also typical examples that can be explained by the recruitment hypothesis (Fig. 1a)[28]. Based on these insights, the TCA cycle and L-lysine biosynthesis were both proposed to have evolved from ubiquitous L-leucine biosynthesis[29]. Although this evolutionary scenario does not directly extend to C3LHyp metabolism and isoprenoid biosynthesis, AcnX shows relatively high structure-based sequence identities with IPMI and HACN (Supplementary Fig. 3). Among other Acn enzymes, only IPMI and HACN from methanogen possessed the "fourth cysteine residue" neighbors Cys273 and Cys110(L) in AtAcnX and TkAcnX (Fig. 2a)[30]. In the crystal structure of the large subunit of IPMI from *Methanococcus jannaschii* only, these four cysteine residues formed two disulfide bonds (Fig. 4i). Conversely, the evolutionary scheme of clostridial ferredoxin included the loss of one of the two [4Fe-4S] clusters and the occasional incorporation of a disulfide bond[18]. On the other hand, the cubane-type [3Fe-4S] cluster in mAcn may convert the linear cluster (similar to the [2Fe-2S] cluster). The "four cysteine ligands" consisted of Cys250, Cys257, Cys421, and Cys424 (but not Cys358); the two formers had now become ones[12]. Therefore, the [4Fe-4S] cluster may have been acquired through random mutagenesis and selection in other Acn enzymes, similar to the AcnX subfamily.

## Methods

**Plasmid construction**. The primer sequences used in the present study are shown in Supplementary Table 2. The AtAcnX gene (Atu4683) from *A. tumefaciens* NBRC 15193 was previously cloned into pQE-80L (Qiagen), by which 11 additional residues (MRGSHHHHHHG) were attached at its N terminus (pHisAtAcnX)[15]. On the other hand, the pETDuet-1 vector (Novagen), with two multiple cloning sites (MCS-1 and MCS-2), was used for the cloning and expression of the TkAcnX protein. The TkAcnX$_S$ (TK_RS06160) and TkAcnX$_L$ (TK_RS06165) genes were amplified by PCR using primers containing appropriate restriction enzyme sites at the 5′- and 3′-ends and the genome DNA of *T. kodakarensis* KOD1 as a template, and were introduced into MCS-1 (*Bam*HI-*Hind*III site) and MCS-2 (*Nde*I-*Xho*I site), respectively, by which 14 additional residues (MGSSHHHHHHSQDP) were attached at the N terminus of TkAcnX$_S$ (pHisTkAcnX). Regarding crystallization, the (underlined) rhinovirus 3 C protease cleavage site from humans was inserted between the (His)$_6$-tag and TkAcnX$_S$ gene in the pHisTkAcnX plasmid by inverse PCR; MGSSHHHHHHSQDP<u>LEVLFQGP</u> (pHisTkAcnX3C). Each plasmid of pHisAtAcnX and pHisTkAcnX3C was transformed into *E. coli* BL21(DE3) and BL21-CodonPlus(DE3)-RIL cells (Novagen), respectively. A site-directed mutation was introduced into the AtAcnX gene by a single round of PCR using sense and antisense primers and pHisAtAcnX as a template. Mutant proteins were expressed and purified by the same procedures as the wild-type enzyme.

**Protein overexpression and purification**. Recombinant *E. coli* cells harboring the constructed plasmids were grown at 37 °C to a turbidity of 0.6 at 600 nm in LB medium containing ampicillin (50 mg/l). After the addition of 1 mM isopropyl-β-D-thiogalactopyranoside, the culture was grown at 18°C for a further 18 h to induce the expression of the respective (His)$_6$-tagged protein. Cells were harvested and resuspended in Buffer A (50 mM sodium phosphate buffer (pH 8.0) containing 300 mM NaCl and 10 mM imidazole). Cells were then disrupted by sonication and the solution was centrifuged. The supernatant was loaded onto a Ni-NTA Superflow column (Qiagen). The column was washed with buffer B (50 mM sodium phosphate buffer (pH 8.0) containing 300 mM NaCl, 10% (v/v) glycerol, and 25 mM imidazole). The enzymes were then eluted with Buffer C (pH 8.0, buffer B containing 250 mM instead of 25 mM imidazole). Regarding the crystallization of TkAcnX, the (His)$_6$-tag was cleaved off with rhinovirus 3 C protease (GE Healthcare), and the proteins obtained were loaded onto a HiLoad 16/600 Superdex 200 pg column (GE Healthcare) equilibrated with Buffer D (20 mM Tris-HCl (pH 8.0) containing 150 mM NaCl). The main single-peak fractions were collected and concentrated by ultrafiltration with Amicon Ultra-15 (Millipore). The SeMet-labeled AtAcnX protein was prepared from *E. coli* BL21(DE3) cells and cultured in minimal media containing 25 mg/ml SeMet. The preparation of recombinant (His)$_6$-tagged ApAcnX and AraC is described in "Supplementary methods".

**UV spectroscopy**. The purified enzyme was diluted with Buffer D, and loaded into a quartz cuvette. The UV visible absorption spectrum was recorded by using a Shimadzu UV-1800 spectrophotometer (Shimadzu GLC Ltd., Tokyo, Japan) at wavelengths of 300–600 nm against Buffer D.

**EPR spectroscopy**. Continuous-wave EPR spectra were obtained on a JEOL TE-300 X-band spectrometer operating with a 100-kHz field modulation. A temperature-dependent analysis was performed at 20 K using an LTR-3 liquid helium cryostat (Air Products). The experimental parameters used to acquire the spectra in Fig. 5 were as follows: microwave frequency; 8.9919–9.0012 GHz monitored by an internal frequency counter, microwave power; 5.0 mW, 100 kHz field modulation magnitude; 0.40–1.25 mT, center field; 280 ± 250 mT, sweep time; 4.0 min, time constant; 0.1 sec, and receiver amplitude; 160–1000. The *g* values of the paramagnetic species were assessed using Li-tetracyanoquinodimethane powder (*g* = 2.0025) as an external standard. Magnetic field strength was calibrated using the hyperfine coupling constant (8.69 mT) of the Mn(II) ion-doped in MgO powder. The (aerobically) purified enzyme (~40 mg/ml) was dialyzed in 50 mM potassium phosphate (pH 7.0) containing 1 mM (NH$_4$)$_2$Fe(SO$_4$)$_2$·6H$_2$O, 10 mM DTT, and 50% (v/v) glycerol under anaerobic conditions using the AnaeroPack system (Mitsubishi Gas Chemical).

**Enzyme assay**. The activity of AtAcnX as a C3LHyp dehydratase was assayed spectrophotometrically in the coupling system with NADH-dependent Pyr2C reductase[15].

**Crystallization**. All crystallization trials were performed at 20°C using the sitting-drop vapor diffusion method. In this method, drops (0.5 μL) of ~20 mg/ml protein in Buffer D were mixed with equal amounts of reservoir solution (as follows) and equilibrated against 70 μL of the same reservoir solution by vapor diffusion: for AtAcnX (native or SeMet-labeled apo-form of the wild-type enzyme, the S449C/C510V mutant, and the C3LHyp-bound form of the S70A mutant), 100 mM Bis-Tris (pH 5.7–6.0), 25–31% (w/v) PEG 3350, and 200 mM MgCl$_2$; for TkAcnX (apo-form), 200 mM ammonium citrate tribasic (pH 7.0) and 24% (w/v) PEG 3350; for TkAcnX (MVA5P-bound form), 50 mM ammonium sulfate, 50 mM Bis-Tris

(pH 6.5), and 30% pentaerythritol ethoxylate (PEE) (15/4 EO/OH). Regarding the crystallization of AtAcnX and TkAcnX in complex with substrates, proteins were prepared in Buffer D containing 40 mM C3LHyp or MVA5P, respectively.

All AtAcnX crystals were soaked in reservoir solution supplemented with 10% (v/v) glycerol. Crystals of the apo-form of TkAcnX were soaked in a reservoir solution containing 35% (w/v) instead of 24% (w/v) PEG 3350. Crystals were then mounted on a nylon loop, flash-cooled, and kept in a stream of nitrogen gas at 100 K during data collection. On the other hand, crystals of TkAcnX in complex with MVA5P were directly mounted onto a nylon loop because the concentration of PEE in the reservoir solution was sufficient as a cryoprotectant.

**X-ray crystallography**. All diffraction data were collected at SPring-8 beamline BL38B1, BL41XU, or BL45XU (Table 1) and processed using XDS[31]. The initial phasing of AtAcnX was performed by the SAD method with the peak data of SeMet-labeled crystals. After selenium sites had been identified and the initial phases were calculated using PHENIX AutoSol[32], density modifications and automated model building were performed with PHENIX AutoBuild[33]. The model obtained was transferred to the native crystal data of AtAcnX. The structures of AtAcnX in complex with C3LHyp and the apo-form of the S449C/C510V mutant of AtAcnX and TkAcnX were elucidated by the molecular replacement method with PHASER in the CCP4 suite[34,35], for which the apo-forms of AtAcnX were used as the search model. The structure of TkAcnX in complex with MVA5P was elucidated by the molecular replacement method, for which the apo-form of TkAcnX was used as the search model.

Further model building for all structures was performed manually with COOT[36] and crystallographic refinement with PHENIX[37]. Detailed data collection and processing statistics are shown in Table 1.

**Reporting summary**. Further information on research design is available in the Nature Research Reporting Summary linked to this article.

## Data availability
Coordinates and structural factors have been deposited in the Protein Data Bank, under the accession codes 7CNP (AtAcnX), 7CNQ (AtAcnX S70A mutant with C3LHyp), 7D2R (AtAcnX S449/C510V mutant), 7CNR (TkAcnX), 7CNS (TkAcnX with MVA5P). All other data are available from the corresponding authors upon reasonable request.

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

## Acknowledgements

We would like to thank Dr. Akira Hirata, Ehime University, for the gift of genomic DNA from *T. kodakarensis* KOD1. Our thanks extend to the staff at beamlines BL38B1, BL41XU, and BL45XU of SPring-8, Japan for their support with data collection.

## Author contributions

S.W. conceived, designed, and partially performed the experiments, and mainly wrote the manuscript. Y.M. performed the X-ray crystallographic analysis and partially wrote the manuscript. Y.W. contributed to the X-ray crystallographic analysis. Y.S. and K.T. contributed to the EPR analysis.

## Competing interests

The authors declare no competing interest.

## Additional information

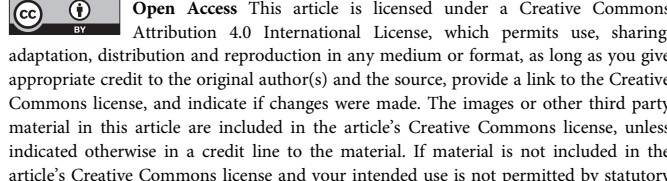

