## [Peer Review File · Communications Biology]

Reviewers' comments:

Reviewer #1 (Remarks to the Author):

This paper structurally characterizes two members of a new subfamily of the well-known aconitase family of enzymes. The most significant part of the work is the likely establishment of an earlier precursor in the evolution of the aconitase family. As such, it is likely of interest to a broad group of researchers. The manuscript is detailed, the work appears technically sound, and the arguments favoring a new common ancestor for this family appear sound.

Reviewer #2 (Remarks to the Author):

The manuscript describes structural characterization of AcnX Type-I and Type-II from *A. tumefaciens* and *T. kodakarensis* containing [2Fe-2S] and [3Fe-4S] cofactors at the active site of these enzymes, respectively.

1) Authors show here crystal structures of [2Fe-2S] cluster bound crystal structures of AtAcnX in the absence and presence of cis-3-Hydroxy-L-proline (C3LHyP) substrate. Based on previously characterized [2Fe-2S] cluster containing dehydratases, authors propose reaction mechanism for conversion of C3LHyP to Pyrroline-2-carboxylate at the [2Fe-2S] active site in AtAcnX. Authors say earlier studies on *Pseudomonas* AcnX had incorrectly assigned iron center and the crystal structure of AtAcnX showing [2Fe-2S] cluster is the correct assignment. I agree that the crystal structure of this putatively more stable AtAcnX likely represent correct active site iron cluster, I believe it would still be highly useful to perform EPR characterization of the AtAcnX iron sulfur cluster (under dithionite reduced and/or oxidized conditions). This would provide another evidence for in solution characterization of the iron center in AtAcnX and would likely shed light on the differences observed from previously characterized *Pseudomonas* AcnX which was characterized as mononuclear iron site by Watanabe et al.

2) In addition, it would be useful to the audience if authors could include UV-Vis spectroscopic data on the enzymes (AtAcnX and TkAcnX) studied in this manuscript and compare with previously reported UV-Vis spectral features of Fe-S clusters.

3) Although the structures of the TkAcnX with and without MVA5P are quite interesting, I believe this part of the study requires more functional and spectroscopic characterization of the enzyme. Authors should also clearly state in the abstract and elsewhere in the manuscript that the crystal structure of the TkAcnX in the 3Fe-4S bound form is the inactive form of the enzyme.

4) Authors state on page 5 (line 28-29) that they observed brown color for the TkAcnX enzyme but no activity. I believe this statement justifies that authors need to characterize this enzyme further as correlating activity just to the observed color in an Eppendorf tube is not sufficient. I would strongly suggest authors to at least perform more thorough spectroscopic characterization before coming to the conclusion. Have authors tried other substrates to test the enzymatic activity for TkAcnX?

5) Are there any sequence similarities or differences near the active site for TkAcnX (this study) and previously characterized *Aeropyrum pernix* AcnX (ApAcnX) (Hayakawa et al, PNAS, 2018) that could suggest observed loss of activity and loss of iron from the cluster? It would be helpful to include a sequence alignment for TkAcnX and ApAcnX since that has been functionally characterized.

6) Loss of active site iron in [4Fe-4S] cluster and conversion to inactive to [3Fe-4S] has been reported under oxidative conditions leading to loss of activity in aconitases. Authors say they tried reconstituting the enzyme with iron anaerobically with no luck. Considering *Thermococcus kodakarensis* being obligate anaerobe, did authors try purifying the enzyme in an anaerobic chamber

starting from the cell lysis and/or simultaneously reconstituting the Fe-S cluster in an anaerobic chamber?

7) Did authors test the enzymatic activity in the cell lysate or supernatant before purification to see if the iron loss occurs during purification steps? I believe testing activity throughout the purification steps will be an important step.

8) Also, it would be worth trying to see if IscU [Fe-S] cluster scaffolding protein can transfer the iron sulfur cluster to the TkAcnX and help reconstitute the enzymatic activity if any present.

9) Although it might be out of scope for this study but it would be still nice to see if authors could fuse domain 4 (small subunit) of TkAcnX with rest of the domains (large subunit) using a linker and expressing it as a single polypeptide chain to see if it would be better behaved in terms of retaining the [4Fe-4S] cluster and the activity.

10) It is unclear if authors tried TkAcnX purification in the presence of the substrate or an analog to engage the labile iron to likely prevent iron loss and conversion from [4Fe-4S] to [3Fe-4S]?

11) Page 11, line 38, caption for fig 1b should read "box" instead of "dox"

Reviewer #3 (Remarks to the Author):

In this paper by Watanabe et al, the authors obtained high-resolution crystal structures of two distinct members of the Aconitase X subfamily, both in the apo and holo forms. The solved structures in combination with sequence analysis were employed to propose a novel evolutionary route for the whole aconitase superfamily. While the crystal structures are new, it is unclear to this reviewer if the authors' results will benefit the field significantly. Indeed, the paper is mainly focused on the description of new crystal structures. Their study falls short of the current state of the art in the field. Without additional evidence, the proposed mechanistic/evolutionary insights seem hypothetical. Some parts of the text are ambiguous and/or difficult to follow (a few of them listed below as minor comments). The manuscript should be carefully revised to improve grammar and readability. Here are my specific concerns that can help the authors preparing an improved version of their manuscript:

Major remarks.

1. The title does not reflect the authors' claims. The authors simply mentioned that they crystallized two proteins which alone won't merit publication in many journals, certainly not in the journal they are targeting. A better title would be something along the lines of: "Crystal structures of aconitase X as cis-3-hydroxy-L-proline dehydratase and mevalonate-5-phosphate dehydratase suggest the molecular evolution of the aconitase superfamily".

2. In the abstract, they say that "these insights will allow us to rewrite the evolutionary scenario ...". The use of the future tense is unclear. The claim seems overstated since apparently, they didn't "rewrite" the whole phylogenetic tree of the aconitase superfamily. I believe what the authors propose is a common ancestor for the aconitase superfamily.

3. The bioinformatics methods (phylogenetic trees of Figures 1C and 1D) must be explained and detailed in the Methods section (only brief descriptions are available in the figure legend). It is unclear whether the authors used ONLY sequence information or also 3D structural information. Assuming that they utilized crystallographic data, figures 1C and 1D should go to a new Figure 7 since they are showing a major conclusion of the paper. On the contrary, if they used only sequence information then they cannot claim that the crystal structures allowed them to gain phylogenetic insights.

4. The authors discuss their results considering only the recruitment hypothesis proposed by Jensen in 1976. Other hypotheses have been postulated since then. Rather than contraposing his results to just Jensen's paper, the authors should consider citing and discussing other relevant literature.

5. The iron-sulfur clusters, which are central to the discussion of their results, have not been

characterized in sufficient detail. In particular, the authors mention several times that the enzymes are brown in color. The authors could complement their crystallographic results with spectroscopy techniques. UV/Vis spectroscopy is the obvious choice together with the mentioned EPR, MB, and ENDOR spectroscopies (page 2). In addition, it is stated that "the molecular evolution of the [Fe-S] cluster within the AcnX subfamily remains unclear" (page 7 line 43): Can the author at least provide some hypothesis/ideas?

6. Related to the above. It is unclear from the text whether the [3Fe-4S] cluster found in TkAcnX is physiologically relevant or just an artifact due to the employed expression/purification/crystallization conditions. It seems that the authors consider such a cluster as an "inactive" form of the enzyme. What is the basis for such assumption? Please explain and/or cite supporting references. Moreover, how did the authors model the hypothetically "active" form of TkAcnX ([4Fe-4S]) (figure 4g). To clarify this issue, the authors should attempt to crystallize the enzyme under anaerobic conditions.

7. It is unclear whether the catalytic mechanism explained on page 5 (lines 6-17) is a fact or a proposed mechanism. Either add the missing references or be more cautious in writing. In particular, how do the authors know that the Fe² atom exists in the Fe³⁺ oxidation state?

8. The paper would benefit from kinetic analysis (reporting not only relative activity as shown in Figure S4 but also other important kinetic parameters like k_{cat} and k_m) of the studied proteins. Specifically, it is stated that "no MVA5P dehydratase activity was observed even in TkAcnX prepares by similar methods" (page 6 line 10), and that the mutant S449C/C510V "was completely inactive" (page 7 line 14). Can the authors show such data?

Minor remarks.

9. Page 1 line 45: Do the authors mean "Of the 20 active site residues including ...".

10. Page 2 line 7: I guess they want to cite the papers and not just the dates.

11. Page 2 lines 25-27. This sentence seems more appropriate for the discussion section.

12. Page 7 line 18: Rather than writing "slightly tremulous" I would express it as "less fluctuating". To make the statement more quantitative, they may want to cite the B-factors.

13. Page 8 line 26: I wouldn't use the expression "trial and error" but rather "random mutagenesis and selection".

14. Page 12 line 38: This sentence is purely hypothetical and should be mentioned.

Although point-by-point responses to the referees are described below, we carried out three additional experiments.

1. To obtain another evidence for in solution characterization of the [Fe-S] cluster in AtAcnX, Reviewers #2 and #3 recommended to carry out UV and ESR spectroscopy analysis. For this purpose, we further prepared two enzymes. First, AcnX from *Pseudomonas* sp. NBRC 111117 (PsAcnX) is the same AcnX_{Type I} enzyme as AtAcnX, and functions as a C3LHyp dehydratase. PsAcnX shows higher sequence similarity to (earlier studied) *Pseudomonas aeruginosa* AcnX (63% of identity) than AtAcnX (46%). Second, L-arabinonate dehydratase (AraC) from *Herbaspirillum huttiense* belongs to different group from Acn protein superfamily, and has the similar scaffold in the [2Fe-2S] cluster coordination to AtAcnX. Therefore, we analyzed not only AtAcnX but also PsAcnX and AraC for UV and ESR spectroscopy analysis. When compared with earlier study, the (more concentrated) enzyme(s) was freshly prepared under partial anaerobic conditions. As results, we obtained that both AtAcnX and PsAcnX surely contain a [2Fe-2S] cluster. The corresponding data is shown as Figure 5. Prof. Kunihiko Tajima and Dr. Yasuhiro Sakurai contributed to ESR analysis significantly, and were newly added to the co-authors.

2. Reviewer #3 recommended to determine the kinetic parameters of AtAcnX mutants. Therefore, among the constructed mutants, four W35A, T72A, I206A, and S293A mutants were subjected to further kinetic analysis with C3LHyp, and the determined parameters are shown as Table S1. The k_{cat}/K_m values of them were also reduced by 1~4 orders of magnitude from the wild-type enzyme, from which these site-directed mutagenic analyses were consistent with the structural insights.

3. Reviewers #2 and #3 recommended to attempt whether active TkAcnX is prepared nor not. Unfortunately, we couldn't purify TkAcnX using anaerobic chamber. Therefore, we originally prepared recombinant (His)₆-tagged ApAcnX. Expectedly (and unfortunately), since the purified ApAcnX was inactive, and the activity was not recovered by the same method as TkAcnX, it was likely that AcnX_{Type-II} enzyme(s) must absolutely purify under anaerobic conditions. Next, we carried out UV and ESR spectroscopy analysis, from which no significant evidence of the presence of the [4Fe-4S] cluster was obtained; these data were shown in Fig. S7.

Point-by-point responses to the referees are as follows.

Reviewer #2

•Authors say earlier studies on *Pseudomonas AcnX* had incorrectly assigned iron center and the crystal structure of *AtAcnX* showing [2Fe-2S] cluster is the correct assignment. I agree that the crystal structure of this putatively more stable *AtAcnX* likely represent correct active site iron cluster, I believe it would still be highly useful to perform EPR characterization of the *AtAcnX* iron sulfur cluster (under dithionite reduced and/or oxidized conditions). This would provide another evidence for in solution characterization of the iron center in *AtAcnX* and would likely shed light on the differences observed from previously characterized *Pseudomonas AcnX* which was characterized as mononuclear iron site by Watanabe et al.

•In addition, it would be useful to the audience if authors could include UV-Vis spectroscopic data on the enzymes (*AtAcnX* and *TkAcnX*) studied in this manuscript and compare with previously reported UV-Vis spectral features of Fe-S clusters.

[Answer] As described above, we carried out to analyze UV and ESR spectroscopy.

•Although the structures of the *TkAcnX* with and without MVA5P are quite interesting, I believe this part of the study requires more functional and spectroscopic characterization of the enzyme. Authors should also clearly state in the abstract and elsewhere in the manuscript that the crystal structure of the *TkAcnX* in the 3Fe-4S bound form is the inactive form of the enzyme.

•Authors state on page 5 (line 28-29) that they observed brown color for the *TkAcnX* enzyme but no activity. I believe this statement justifies that authors need to characterize this enzyme further as correlating activity just to the observed color in an Eppendorf tube is not sufficient. I would strongly suggest authors to at least perform more thorough spectroscopic characterization before coming to the conclusion. Have authors tried other substrates to test the enzymatic activity for *TkAcnX*?

[Answer] These suggestions are for characterization of *TkAcnX*. As described above, we carried out the UV and ESR spectroscopy of *TkAcnX* (and *ApAcnX*). The UV spectra suggested the presence of (unidentified) [Fe-S] cluster at least. Mevalonate 5P is commercially available, but too expensive to carry out such as experiment.

•Are there any sequence similarities or differences near the active site for *TkAcnX* (this study) and previously characterized *Aeropyrum pernix AcnX* (*ApAcnX*) (Hayakawa et al, PNAS, 2018) that could suggest observed loss of activity and loss of iron from the cluster? It would be helpful to include a sequence alignment for *TkAcnX* and *ApAcnX* since that has been functionally characterized.

[Answer] We added the structural-based alignment between TkAcnX and ApAcnX as Fig. S6.

- Loss of active site iron in [4Fe-4S] cluster and conversion to inactive to [3Fe-4S] has been reported under oxidative conditions leading to loss of activity in aconitases. Authors say they tried reconstituting the enzyme with iron anaerobically with no luck. Considering *Thermococcus kodakarensis* being obligate anaerobe, did authors try purifying the enzyme in an anaerobic chamber starting from the cell lysis and/or simultaneously reconstituting the Fe-S cluster in an anaerobic chamber?

- Did authors test the enzymatic activity in the cell lysate or supernatant before purification to see if the iron loss occurs during purification steps? I believe testing activity throughout the purification steps will be an important step.

- Also, it would be worth trying to see if IscU [Fe-S] cluster scaffolding protein can transfer the iron sulfur cluster to the TkAcnX and help reconstitute the enzymatic activity if any present.

[Answer] Unfortunately, we couldn't purify TkAcnX, and reconstruct the [4Fe-4S] cluster in anaerobic chamber. As described in text, we also unsuccessfully prepared active ApAcnX under the same procedures, suggesting that the purification under anaerobic conditions may be absolutely necessary for AcnX_{Type-II}.

- Although it might be out of scope for this study but it would be still nice to see if authors could fuse domain 4 (small subunit) of TkAcnX with rest of the domains (large subunit) using a linker and expressing it as a single polypeptide chain to see if it would be better behaved in terms of retaining the [4Fe-4S] cluster and the activity.

[Answer] Indeed, to estimate the similar purpose, we have already attempted to the fusion of small and large subunits of (heterodimeric) AcnX_{Type-II} from bacteria without a linker, which corresponds to AcnX_{Type-IIa} as a C3LHyp dehydratase in this study. Rapidly inactivation after (aerobic purification) is also found in this enzyme. As results, the fused protein showed the same activity as (monomeric) AcnX_{Type-I} and AcnX_{Type-IIa}. Although this experiment gave the significant insight for molecular evolution of AcnX, the lability was not improved. These explanations were briefly added in text.

- It is unclear if authors tried TkAcnX purification in the presence of the substrate or an analog to engage the labile iron to likely prevent iron loss and conversion from [4Fe-4S] to [3Fe-4S]?

[Answer] The substrate is too expensive to add in purification buffers.

•Page 11, line 38, caption for fig 1b should read “box” instead of “dox”

[Answer] This was appropriately corrected.

Reviewer #3

•The title does not reflect the authors’ claims. The authors simply mentioned that they crystallized two proteins which alone won’t merit publication in many journals, certainly not in the journal they are targeting. A better title would be something along the lines of: “Crystal structures of aconitase X as cis-3-hydroxy-L-proline dehydratase and mevalonate-5-phosphate dehydratase suggest the molecular evolution of the aconitase superfamily”.

[Answer] The title was appropriately corrected.

•In the abstract, they say that “these insights will allow us to rewrite the evolutionary scenario ...”. The use of the future tense is unclear. The claim seems overstated since apparently, they didn’t “rewrite” the whole phylogenetic tree of the aconitase superfamily. I believe what the authors propose is a common ancestor for the aconitase superfamily.

[Answer] This sentence was modified as follows.

These insights will give novel insights for the evolutionary scenario of the aconitase superfamily based on the recruitment hypothesis.

•The bioinformatics methods (phylogenetic trees of Figures 1C and 1D) must be explained and detailed in the Methods section (only brief descriptions are available in the figure legend). It is unclear whether the authors used ONLY sequence information or also 3D structural information. Assuming that they utilized crystallographic data, figures 1C and 1D should go to a new Figure 7 since they are showing a major conclusion of the paper. On the contrary, if they used only sequence information then they cannot claim that the crystal structures allowed them to gain phylogenetic insights.

[Answer] First, phylogenetic tree (Fig. 1C) was constructed based on sequence identity of domain 4, in National Center for Biotechnology Information (NCBI). This sentence was added to the figure legend. Phylogenetic tree of AcnX subfamily was originally constructed based on the sequence similarity. This sentence was added to the figure legend. Even if AcnX structure is available in this study, it may be very difficult to construct the “structural-based” phylogenetic tree between AncX and other aconitase enzymes, due to large structural differences.

•The authors discuss their results considering only the recruitment hypothesis proposed by Jensen in 1976. Other hypotheses have been postulated since then. Rather than contraposing his results to just Jensen's paper, the authors should consider citing and discussing other relevant literature.

[Answer] Other evolutionary hypothesis was discussed as follows.

The β -decarboxylating dehydrogenases, ICDH, HICDH, and IPMDH, included in the TCA cycle and the L-lysine and -leucine biosynthetic pathways belong to a single protein superfamily, whose relationships are also typical examples that can be explained by the recruitment hypothesis (Fig. 1a) [28]. Based on these insights, the TCA cycle and L-lysine biosynthesis were both proposed to have evolved from ubiquitous L-leucine biosynthesis [29]. Although this evolutionary scenario does not directly extend to C3LHyp metabolism and isoprenoid biosynthesis, AcnX shows relatively high structure-based sequence identities with IPMI and HACN (Fig. S3).

•The iron-sulfur clusters, which are central to the discussion of their results, have not been characterized in sufficient detail. In particular, the authors mention several times that the enzymes are brown in color. The authors could complement their crystallographic results with spectroscopy techniques. UV/Vis spectroscopy is the obvious choice together with the mentioned EPR, MB, and ENDOR spectroscopies (page 2). In addition, it is stated that “the molecular evolution of the [Fe-S] cluster within the AcnX subfamily remains unclear” (page 7 line 43): Can the author at least provide some hypothesis/ideas?

[Answer] As described above, we carried out to analyze UV and ESR spectroscopy. This sentence means that “the molecular evolution of the [Fe-S] cluster between AcnX_{Type-I} and AcnX_{Type-IIa} remains unclear, because the later potentially possessed only two cysteine residues for [Fe-S] cluster-binding sites. The characterization of the S449C/C510V mutant of AtAcnX may be helpful to estimate this question”. This sentence was appropriately modified.

•Related to the above. It is unclear from the text whether the [3Fe-4S] cluster found in TkAcnX is physiologically relevant or just an artifact due to the employed expression/purification/crystallization conditions. It seems that the authors consider such a cluster as an “inactive” form of the enzyme. What is the basis for such assumption? Please explain and/or cite supporting references. Moreover, how did the authors model the hypothetically “active” form of TkAcnX ([4Fe-4S]) (figure 4g). To clarify this issue, the authors should attempt to crystallize the enzyme under anaerobic conditions.

[Answer] After the preparation of anaerobic chamber, we would attempt to crystallize the active form of TkAcnX.

•It is unclear whether the catalytic mechanism explained on page 5 (lines 6-17) is a fact or a proposed mechanism. Either add the missing references or be more cautious in writing. In particular, how do the authors know that the Fe²⁺ atom exists in the Fe³⁺ oxidation state?

[Answer] We further discussed the putative catalytic mechanism in more detail. EPR spectroscopy suggest clearly the Fe(III) oxidation state.

•The paper would benefit from kinetic analysis (reporting not only relative activity as shown in Figure S4 but also other important kinetic parameters like k_{cat} and K_m) of the studied proteins. Specifically, it is stated that “no MVA5P dehydratase activity was observed even in TkAcnX prepares by similar methods” (page 6 line 10), and that the mutant S449C/C510V “was completely inactive” (page 7 line 14). Can the authors show such data?

[Answer] As described above, we determined kinetic parameters of several AtAcnX, and added as Table S1.

Minor remarks.

•Page 1 line 45: Do the authors mean “Of the 20 active site residues including ...”.

[Answer] This was appropriately corrected.

•Page 2 line 7: I guess they want to cite the papers and not just the dates.

[Answer] This was appropriately corrected.

•Page 2 lines 25-27. This sentence seems more appropriate for the discussion section.

[Answer] This was appropriately modified as follows.

Furthermore, based on EPR and site-directed mutagenic analysis, a mononuclear Fe(III) center may be coordinated with one glutamate and two cysteine residues, (with orange-, light green-, and green-colored circles in Fig. 2b, respectively).

•Page 7 line 18: Rather than writing “slightly tremulous” I would express it as “less fluctuating”. To make the statement more quantitative, they may want to cite the B-factors.

[Answer] This was appropriately modified to the sentence including B-factor values as follows.

An anomalous difference Fourier map showed that this mutant enzyme had the [2Fe-2S] cluster (Fig. 4c). On the other hand, the [2Fe-2S] cluster may fluctuate less for the following reasons: 1) the *B*-factor values of the Fe1, Fe2, S1, and S2 atoms (43.98, 23.19, 29.57, and 22.66, respectively, for chain A) were higher than those in the wild-type enzyme (10.49, 9.11, 10.18, and 10.49, respectively, for chain A); 2) the anomalous difference peak in the Fe1 atom was significantly smaller than that in the Fe2 atom; and 3) the *B*-factor value of the Fe2 atom was higher than that of the Fe1 atom.

•Page 8 line 26: I wouldn't use the expression "trial and error" but rather "random mutagenesis and selection".

[Answer] This was appropriately corrected.

•Page 12 line 38: This sentence is purely hypothetical and should be mentioned.

[Answer] A word of "hypothetical" was added in Fig. 4g legend.

Finally, the manuscript was optimized by a native speaker again.

Reviewers' comments:

Reviewer #2 (Remarks to the Author):

In the revised manuscript, authors have now provided UV and EPR spectroscopy data supporting some of their observations. Author's response regarding not being able to perform anaerobic purification of TkAcnX and substrate being expensive for activity studies are not satisfactory. Collaborative efforts should have been taken to come up with anaerobic purification of TkAcnX in order to gain insights into the nature of the active site of the enzyme.

Reviewer #3 (Remarks to the Author):

I am glad to see that the authors made a great effort to improve the quality of their work and I believe now the manuscript is essentially ready for publication. The UV/Visible and EPR spectra are particularly informative and help clarifying the nature of the iron-sulfur cluster. Similarly, it is encouraging to see that the structural and functional (kinetic analysis) studies are self-consistent. It is a pity that the authors were not able to obtain active TkAcnX and I am sure that they will keep trying to purify the enzyme under anaerobic conditions in follow-up publications. The only minor comment is that I would add a paragraph in the Methods section about how the UV/Vis spectra were taken. Additionally, it seems that spectra are saturated (absorbance larger than 1) after addition of sodium dithionite. In order to visualize the full spectra, I would recommend the authors to either dilute the samples or use a shorter path-length cuvette. Moreover, the UV/Vis spectra of TkAcnX (Fig. S7) seem to deviate upwards at lower wavelengths, possibly suggesting protein aggregation. In the text, AtAcnX is stated to be monomeric (line 101, page 3). However, the oligomeric state of TkAcnX is not mentioned. I wonder if the observed inactivity of TkAcnX may arise not only because of the [3Fe-4S] cluster but also because of oligomerization/aggregation in solution. The authors may briefly comment on this topic if they consider appropriate.

Based on suggestions by referees, we briefly modified the manuscript as yellow highlighters.

1. A paragraph about UV spectroscopy was added in the Methods section (p11, L4-7).
2. We had already recorded the diluted sample of $\text{Na}_2\text{S}_2\text{O}_4$ reduction. The corresponding data was added in Fig. 5a, c, e and Fig. S7a, b (and the legends).
3. The insight for subunit assembly of TkAcnX (as a heterodimeric structure) was added (p6, L18-19).
4. Since no significant peak in void volume was observed during the gel-filtration for (aerobic) purification of TkAcnX, it is likely that the inactivation is not due to protein aggregation (data not shown). This sentence was added (p7, L10-12).